# The effects of caloric restriction on adipose tissue and metabolic health are sex- and age-dependent

Karla J Suchacki[1†], Benjamin J Thomas[1†], Yoshiko M Ikushima[1†], Kuan-Chan Chen[1,2], Claire Fyfe[3], Adriana AS Tavares[1,4], Richard J Sulston[1], Andrea Lovdel[1], Holly J Woodward[1], Xuan Han[1], Domenico Mattiucci[1], Eleanor J Brain[1], Carlos J Alcaide-Corral[1,4], Hiroshi Kobayashi[5], Gillian A Gray[1], Phillip D Whitfield[6,7], Roland H Stimson[1], Nicholas M Morton[1], Alexandra M Johnstone[3], William P Cawthorn[1]*

[1]University/BHF Centre for Cardiovascular Science, University of Edinburgh, The Queen's Medical Research Institute, Edinburgh BioQuarter, Edinburgh, United Kingdom; [2]Division of Endocrinology and Metabolism, Department of Internal Medicine, Tri-Service General Hospital, National Defense Medical Center, Taipei, Taiwan; [3]Rowett Institute, University of Aberdeen, Aberdeen, United Kingdom; [4]Edinburgh Imaging, University of Edinburgh, The Queen's Medical Research Institute, Edinburgh BioQuarter, Edinburgh, United Kingdom; [5]Department of Stem Cell Biology, Research Institute, National Center for Global Health and Medicine, Tokyo, Japan; [6]Division of Biomedical Sciences, University of the Highlands and Islands, Centre for Health Sciences, Inverness, United Kingdom; [7]Institute of Infection, Immunity & Inflammation, University of Glasgow, Glasgow, United Kingdom

*For correspondence:
W.Cawthorn@ed.ac.uk

†These authors contributed equally to this work

**Abstract** Caloric restriction (CR) reduces the risk of age-related diseases in numerous species, including humans. CR's metabolic effects, including decreased adiposity and improved insulin sensitivity, are important for its broader health benefits; however, the extent and basis of sex differences in CR's health benefits are unknown. We found that 30% CR in young (3-month-old) male mice decreased fat mass and improved glucose tolerance and insulin sensitivity, whereas these effects were blunted or absent in young females. Females' resistance to fat loss was associated with decreased lipolysis, energy expenditure and fatty acid oxidation, and increased postprandial lipogenesis, compared to males. The sex differences in glucose homeostasis were not associated with differential glucose uptake but with altered hepatic ceramide content and substrate metabolism: compared to CR males, CR females had lower TCA cycle activity and higher blood ketone concentrations, a marker of hepatic acetyl-CoA content. This suggests that males use hepatic acetyl-CoA for the TCA cycle whereas in females it accumulates, stimulating gluconeogenesis and limiting hypoglycaemia during CR. In aged mice (18-months old), when females are anoestrus, CR decreased fat mass and improved glucose homeostasis similarly in both sexes. Finally, in a cohort of overweight and obese humans, CR-induced fat loss was also sex- and age-dependent: younger females (<45 years) resisted fat loss compared to younger males while in older subjects (>45 years) this sex difference was absent. Collectively, these studies identify age-dependent sex differences in the metabolic effects of CR and highlight adipose tissue, the liver and oestrogen as key determinants of CR's metabolic benefits. These findings have important implications for understanding the interplay between diet and health, and for maximising the benefits of CR in humans.

## Editor's evaluation

This manuscript breaks new ground in old soil; i.e. sex differences in mouse and human studies and one of the greatest challenges facing translational investigators is the remarkable difference in phenotypes by sex. Defining those differences has been relatively straightforward, but understanding the underlying basis for the phenotypic effects continues to be difficult. This paper demonstrates the sex-and age-dependent effects of calorie restriction on adipose tissue and body composition. It then goes further in describing the metabolic distinctions that arise between males and females. As such, this study will open up new opportunities to explore these sex differences and to test the hypothesis that estrogen in younger animals and humans may be at the center of these effects. Furthermore, it provides a basis for determining why weight loss may be difficult for some individuals with calorie-restricted diets.

## Introduction

Caloric restriction (CR) is a therapeutic nutritional intervention involving a sustained decrease in calorie intake whilst maintaining adequate nutrition. CR extends lifespan and reduces the risk of age-related diseases in numerous species, ranging from yeast to primates (*Lin et al., 2002*; *Weindruch et al., 1986*; *Mattison et al., 2017*). CR can have detrimental effects, including bone loss (*Villareal et al., 2016*) and increased susceptibility to infections (*Speakman and Mitchell, 2011*), and therefore it may not be suitable for all individuals; however, trials of CR in humans show numerous health benefits, including the prevention of cardiovascular disease, hypertension, obesity, type 2 diabetes, chronic inflammation, and risk of certain cancers (*Most et al., 2017*). Thus, the ability of CR to promote healthy ageing is now also recognised in humans. In addition to these health benefits, many effects of CR reflect evolutionary adaptations that confer a survival advantage during periods of food scarcity (*Speakman and Mitchell, 2011*). Establishing the extent and basis of CR's effects may thereby reveal new knowledge of healthy ageing and the interplay between food, nutrition, and health.

A key contributor to the health benefits of CR is its impact on metabolic function. Ageing is characterised by hepatic insulin resistance, hyperinsulinaemia, and excessive accumulation of white adipose tissue (WAT), particularly visceral WAT (*Barzilai et al., 1998*; *López-Otín et al., 2016*; *Mancuso and Bouchard, 2019*). The latter is coupled with adipose dysfunction, whereby WAT becomes unable to meet the demands for safe lipid storage. This results in ectopic lipid accumulation in the liver and other tissues, contributing to insulin resistance and metabolic dysregulation (*Barzilai et al., 1998*; *Mancuso and Bouchard, 2019*). CR counteracts these effects, decreasing WAT mass, increasing fatty acid (FA) oxidation, preventing ectopic lipid deposition and enhancing insulin sensitivity and glucose tolerance (*Speakman and Mitchell, 2011*). Notably, studies in rodents show that removal of visceral WAT, independent of CR, is sufficient to prevent insulin resistance, improve glucose tolerance and increase lifespan (*Gabriely et al., 2002*; *Muzumdar et al., 2008*; *Tran et al., 2008*). Visceral WAT loss in humans is also strongly associated with CR's metabolic benefits (*Larson-Meyer et al., 2006*). Thus, the ability of CR to decrease visceral WAT, and the resultant improvements in hepatic insulin sensitivity, are likely central to CR's effects on metabolic function and healthy ageing.

This importance of adiposity and metabolic function raises the possibility of sex differences in the CR response. Indeed, it is now well established that metabolic homeostasis and adipose biology differ substantially between males and females (*Mauvais-Jarvis, 2018*; *Oliva et al., 2020*; *Valencak et al., 2017*; *Maggi and Della Torre, 2018*); however, most preclinical rodent studies have used males only (*Prendergast et al., 2014*), suggesting that much of the CR literature may have overlooked sex as a potential determinant of CR's effects. Nevertheless, some clinical and preclinical CR studies from our lab and others have identified sexually dimorphic responses (*Kane et al., 2018*; *Redman et al., 2007*; *Shi et al., 2007*; *Cawthorn et al., 2016*), whereby males lose fat mass to a greater extent than females. Oestrogens underlie many metabolic sex differences (*Mauvais-Jarvis, 2018*; *Della Torre et al., 2018*), suggesting that oestrogen may contribute to females' resistance to fat loss during CR. However, the extent and basis of sexual dimorphism in CR's metabolic effects remains to be firmly established.

Herein, we first systematically review the CR literature to establish the degree to which sex has been overlooked a determinant of the CR response. To further elucidate the extent and basis of sex differences, we studied CR in male and female mice and humans at ages where oestrogen is

physiologically active or absent. Together, our findings show that the CR field has largely overlooked sex differences and reveal that both mice and humans display age-dependent sexual dimorphism in the metabolic effects of CR.

## Results

### Sex is routinely overlooked as a biological variable in the CR literature

One review of the recent CR literature found that most rodent studies use males only, with females typically used only to address female-specific experimental questions (*Kane et al., 2018*); however, the extent of this bias in earlier CR studies, and whether it applies to human CR research, has not been assessed. To address these issues, we first systematically quantified the use of males and females, and the consideration of sex as a biological variable, in mouse and human CR studies (*Figure 1*, *Figure 1—figure supplement 1*). We focused on research published since 2003, when the European Commission first highlighted the importance of addressing sex as a biological variable (*Lee, 2018*). We also excluded studies that necessarily focused on a single sex, such as those addressing effects of CR on female reproductive function, leaving only studies in which there is no scientific rationale for ignoring potential sexual dimorphism. We found that male-only studies predominated for mice, accounting for ~64% of all mouse CR studies since 2003 (*Figure 1A*, *Figure 1—figure supplement 1B*). This is consistent with previous analyses of the more-recent CR literature (*Kane et al., 2018*). Fewer than 20% of studies used females only, while around 7.3% combined males and females, suggesting an assumption that sex would not influence experimental outcomes. In contrast, ~60% of human studies combined males and females, while ~23% used only females and ~12% only males (*Figure 1C*, *Figure 1—figure supplement 1B*). Strikingly, by the end of 2021, studies that included both sexes and analysed data with sex as a variable were in a minority for both mice (~3.4%) and humans (~4.5%) (*Figure 1A and C*). Moreover, the proportion of studies using each sex or combination of sexes has remained relatively constant since 2003 (*Figure 1A and C*). Thus, efforts to increase the consideration of sex in experimental design, as promoted by the European Commission (*Lee, 2018*), Canadian Institutes of Health Research (CIHR) and National Institutes of Health (NIH) (*Johnson et al., 2014*), appear to have had little impact on the field of CR research.

We next focused on the minority of studies that did consider sex in their experiments ('M&F separate') and that included analysis of metabolic parameters. We found that results in ~74% of mouse studies (*Figure 1B*) and ~62% of human studies (*Figure 1D*) indicated a sex difference in the CR response. Thus, sex differences have been described in the CR literature, but the continuing dearth of studies that include both sexes suggests that this issue continues to be overlooked in the CR field.

### Females resist CR-induced weight loss and fat loss

To explore sex differences in the CR response, we assessed the effects of 30% CR, implemented from 9 to 15 weeks of age, in C57BL/6 J and C57BL/6 N mice. As expected, CR decreased body mass in both males and females (*Figure 2A*) but this effect was greater in males, with ANOVA confirming a significant sex-diet interaction. This sex difference in response to CR is particularly clear when body mass is presented relative to baseline, pre-CR levels (*Figure 2B*).

To determine how fat and lean mass contribute to these diet and sex effects, body composition was assessed weekly using time-domain nuclear magnetic resonance (TD-NMR; *Figure 2C–F*, *Figure 2—figure supplement 1A–B*). CR decreased fat and lean mass in males, whereas females maintained fat mass and lost only lean mass (*Figure 2C–F*). A significant sex-diet interaction was apparent for absolute fat mass (*Figure 2C*) and for fat mass relative to baseline (*Figure 2D*); the latter showed that CR decreased fat mass in males, whereas AL males and AL or CR females increased fat mass to a similar extent over the 6-week duration (*Figure 2D*). For lean mass, a significant sex-diet interaction occurred for absolute mass, with losses being greater in males than in females (*Figure 2E*). However, when compared to baseline lean mass, the CR vs AL effect was similar between the sexes, in part because AL females continued to increase lean mass over time (*Figure 2F*).

Given that these body composition changes coincide with changes in overall body mass, we also assessed fat and lean mass as % body mass. Males preferentially lost fat mass and preserved lean mass in response to CR, with % lean mass being greater in CR vs AL-fed males (*Figure 2—figure supplement 1A–B*). In contrast, diet did not alter % fat or % lean mass in females, indicating that changes in

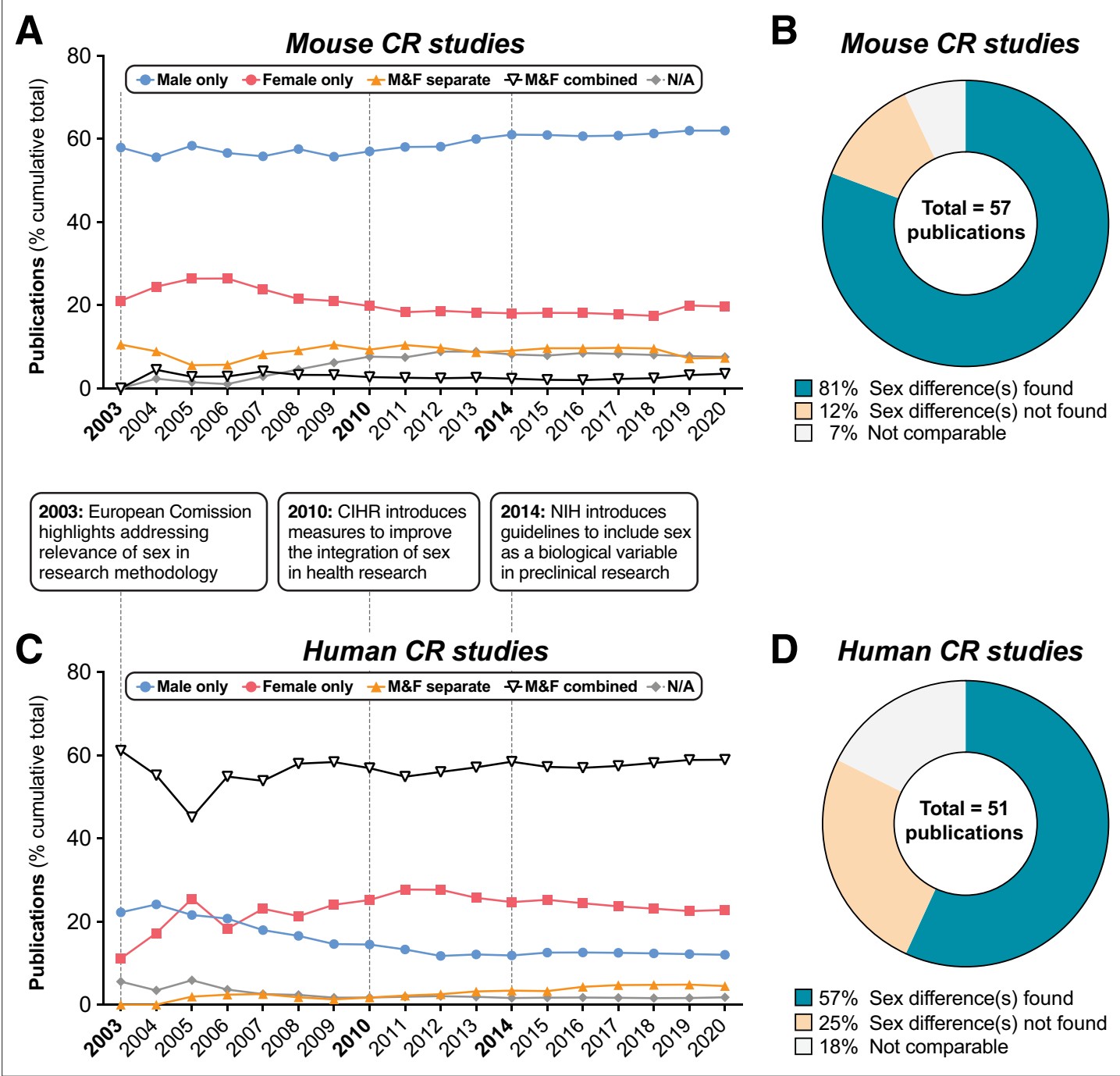

**Figure 1.** Summary of sex differences in mouse and human CR research. PubMed was searched using MeSH terms to identify research articles that studied caloric restriction in vivo in mice (**A,B**) or humans (**C,D**), published between 2003 and 2021. Search results were then classified into the following groups depending on the sexes included in each study: 'Male only'=male subjects used exclusively; 'Female only'=female subjects used exclusively; 'Male & Female separate'=male and female subjects used with data reported respective to each sex, allowing sex differences to be addressed; 'Male & female combined'=male and female subjects used with data from both sexes combined either in part or in full; 'N/A'=no sex data available. (**A,C**) Cumulative publications for studies within each group. The boxes between (**A**) and (**C**) highlight the dates of funders' policies highlighting the importance of sex as a biological variable. (**B,D**) Pie charts of those studies in the 'Male & Female separate' group that considered sex a biological variable, and the proportion of these that did or did not detect sex differences. Source data are provided as a Source Data file. See also *Figure 1—figure supplement 1*.

The online version of this article includes the following source data and figure supplement(s) for figure 1:

**Source data 1.** Literature search to identify sex differences in mouse and human CR research.

*Figure 1 continued on next page*

*Figure 1 continued*

**Figure supplement 1.** Methods used and additional outcomes relating to the comprehensive review of the CR literature.

**Figure supplement 1—source data 1.** Additional outcomes relating to the comprehensive review of the CR literature.

body composition were proportionate to the overall changes in body mass (*Figure 2—figure supplement 1A–B*).

To determine how these changes relate to regional adiposity, we measured adipose depot masses after 6 weeks of AL or CR diet (*Figure 2G*, *Figure 2—figure supplement 1C*, *Figure 2—figure supplement 2A*). We found that in males but not females CR decreased the absolute mass of gonadal (gWAT), inguinal (iWAT), mesenteric (mWAT), and perirenal (pWAT) WAT depots, as well as brown adipose tissue (BAT; *Figure 2—figure supplement 1C*). CR also tended to decrease the absolute mass of pericardial WAT (pcWAT) in males only (*Figure 2—figure supplement 1C*). To determine if these changes were proportionate to changes in overall body mass, we further analysed the mass of each adipose depot as % body mass (*Figure 2G*, *Figure 2—figure supplement 2A*). This showed that the significant sex-diet interaction persisted for gWAT, iWAT, mWAT and pWAT. CR did not affect % BAT mass in either sex, although % BAT mass was significantly greater in males than in females (*Figure 2—figure supplement 2A*). Thus, both in absolute terms and as % body mass, CR decreases WAT mass in males but not in females.

We next analysed the masses of other tissues to determine if this sex-dependent effect of CR is unique to WAT. CR significantly decreased the absolute mass of the liver, pancreas, kidneys, gastrocnemius muscle (gastroc), heart, spleen, and thymus (*Figure 2—figure supplement 2B*). Significant sex-diet effects were detected for the liver and kidney, with CR causing greater decreases in males than in females; however, for each of the other tissues the CR effect was similar between the sexes (*Figure 2—figure supplement 2B*). In contrast to these effects on absolute mass, CR did not significantly affect the relative masses of each of these tissues, nor did sex influence the CR effect (*Figure 2—figure supplement 2C*). This indicates that the absolute masses of these tissues decreased in proportion to the changes in overall body mass. One notable exception is the adrenal glands, the mass of which was increased with CR to a greater extent in males than in females (*Figure 2—figure supplement 2B–C*). Together, these data show that the sex differences in CR-induced loss of body mass and fat mass are driven primarily by decreased WAT mass in males, which females robustly resist.

## Females resist adipocyte hypotrophy and lipolysis during CR

Differences in WAT mass can be driven by changes in adipocyte size and/or adipocyte number. Thus, given the marked sex differences in the effect of CR on WAT mass, we next investigated this effect at the level of adipocyte size (*Figure 2H–I*, *Figure 2—figure supplement 3A–B*). CR significantly decreased average adipocyte area in males but not in females, both for gWAT (*Figure 2H–I*) and for mWAT (*Figure 2—figure supplement 3A–B*). This suggests that females resist lipolysis during CR. Consistent with this, CR increased plasma NEFA concentrations in males but not in females (*Figure 2J*). To further assess adipocyte lipolysis, we analysed phosphorylation of hormone-sensitive lipase (HSL) in iWAT. CR stimulated HSL phosphorylation in males but not in females (*Figure 2K–L*, *Figure 2—figure supplement 3C*). Moreover, across both diets HSL phosphorylation was lower in females whereas total HSL was increased by CR in females only (*Figure 2K–L*, *Figure 2—figure supplement 3C*). Together, these observations show that females resist lipolysis during CR.

## Females suppress energy expenditure and increase postprandial lipogenesis more than males during CR

We next investigated if altered energy expenditure also contributes to the sex differences in the CR response. To do so, we used indirect calorimetry to analyse mice during week 1 and week 3 of CR, corresponding to periods of weight loss and weight maintenance, respectively (*Figure 2A*). This revealed that CR decreased total energy expenditure in both sexes, with greater decreases occurring during week 3 compared to week 1, and during nighttime compared to daytime (*Figure 3A–B*). Notably, during week 1, CR females had lower daytime, nighttime, and total energy expenditure than CR males (*Figure 3A–B*), likely explaining females' resistance to weight loss and fat loss during the first week of CR (*Figure 2A–D*). The daytime diet differences disappeared when normalised to lean body

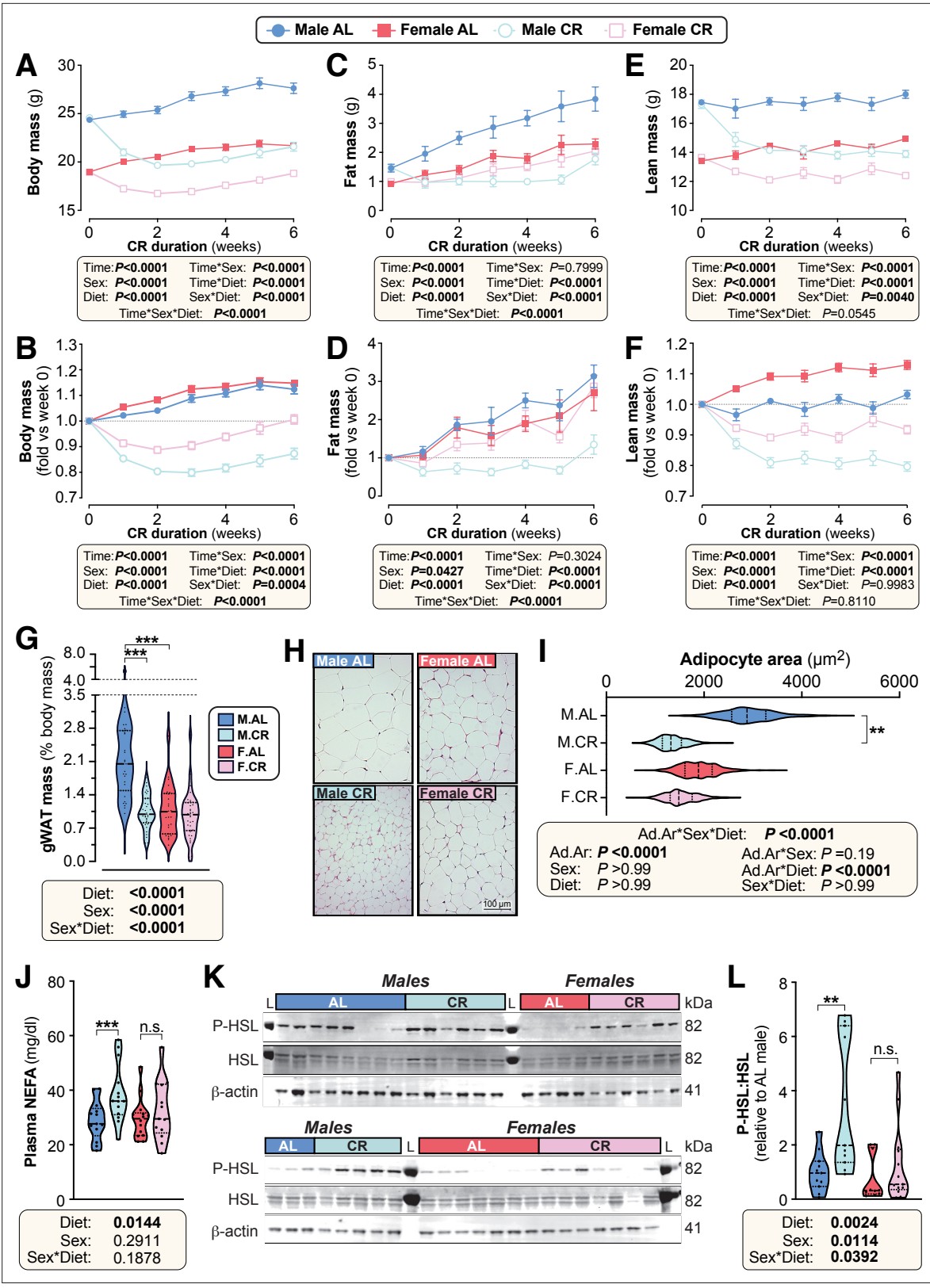

**Figure 2.** Female mice resist weight loss, fat loss, and lipolysis during CR. Male and female mice on a C57BL/6NCrl or C57BL/6 J background were fed ad libitum (AL) or a 30% CR diet from 9 to 15 weeks of age (0–6 weeks of CR). (**A–F**) Each week mice were weighed (**A,B**) and body composition was determined by TD-NMR (**C–F**). Body mass, fat mass, and lean mass are shown as absolute masses (**A,C,E**) or fold-change relative to baseline (**B,D,F**). (**G**) The mass of gWAT (gonadal WAT) was recorded at necropsy and is shown as % body mass. (**H,I**) Micrographs of H&E-stained sections of gWAT (**H**) were

*Figure 2 continued on next page*

*Figure 2 continued*

used for histomorphometric analysis of adipocyte area (**I**); in (**H**), scale bar = 100 μm. (**J**) Plasma was sampled at 15 weeks of age and the concentration of non-esterified fatty acids (NEFA) was assayed. (**K,L**) At 10 weeks of age, during the period of maximal weight and fat loss, a separate cohort of mice were culled and iWAT was collected for analysis of the indicated proteins by Western blotting. Fluorescent Western blots (**K**) were quantified using LICOR software (**L**); L=protein ladder. Data in (**H**) and (**K**) show representative micrographs (**H**) and immunoblots (**K**). Data in (**A–F**) are shown as mean ± SEM of the following numbers of mice per group: *male AL*, n=42; *female AL*, n=43; *male CR*, n=44; *female CR*, n=52. Data in (**G**), (**J**) and (**L**) are shown as violin plots of the following numbers of mice per group: *male AL,* n=29(G), 13 (**J**) or 11 (**L**); *female AL,* n=28(G), 14 (**J**) or 10 (**L**); *male CR*, n=33 (**G**), 14 (**J**) or 11 (**L**); *female CR,* n=34 (**G**) or 13 (**J,L**). Data in (**H–I**) are shown as representative micrographs or violin plots from 5 (*male AL*) or 6 (*female AL, male CR, female CR*) mice per group. For (**A–F**), significant effects of diet, sex and/or time, and interactions thereof, were determined by three-way ANOVA or a mixed-effects model. For (**G**), (**J**) and (**L**), significant effects of diet and/or sex were determined by two-way ANOVA with Tukey's or Šídák's multiple comparisons tests. For (**I**), significant effects of diet and/or sex on adipocyte area (Ad.Ar) were determined using a mixed-effects model, while significant differences in median Ad.Ar between AL and CR mice were determined two-way ANOVA with Šídák's multiple comparisons test. p *Values* from ANOVA or mixed models are shown beneath the graphs, as indicated. For (**G**), (**I**), (**J**) and (**L**), significant differences between comparable groups are indicated by * (p<0.05), ** (p<0.01), or *** (p<0.001). Source data are provided as a Source Data file. See also *Figure 2—figure supplements 1–4*.

The online version of this article includes the following source data and figure supplement(s) for figure 2:

**Source data 1.** Female mice resist weight loss, fat loss, and lipolysis during CR.

**Figure supplement 1.** The effect of CR on percent fat and lean mass differs between male and female mice.

**Figure supplement 1—source data 1.** The effect of CR on percent fat and lean mass differs between male and female mice.

**Figure supplement 2.** CR and sex effects on the masses of adipose and non-adipose tissues.

**Figure supplement 2—source data 1.** CR and sex effects on the masses of adipose and non-adipose tissues.

**Figure supplement 3.** CR decreases adipocyte size and stimulates lipolysis in male but not in female mice.

**Figure supplement 3—source data 1.** CR decreases adipocyte size and stimulates lipolysis in male but not in female mice.

**Figure supplement 4.** Sex differences in the effects of CR on body mass and composition persist when CR mice are fed in the evening.

**Figure supplement 4—source data 1.** Sex differences in the effects of CR on body mass and composition persist when CR mice are fed in the evening.

mass (*Figure 3—figure supplement 1A*), suggesting that they are driven primarily by the loss of lean mass and a consequent reduction in basal metabolic rate. In contrast, CR still decreased in nighttime and total energy expenditure even when normalised to lean body mass (*Figure 3—figure supplement 1A*). The relationship between lean mass and total energy expenditure (*P,* Slope) did not differ among the groups; however, the intercepts of the best-fit lines (*P,* Intercept) did differ significantly, both during week 1 (*Figure 3C*) and week 3 (*Figure 3—figure supplement 1B*). Thus, for a given lean body mass, CR-fed males and females had significantly lower energy expenditure than their AL-fed counterparts (*Figure 3C*, *Figure 3—figure supplement 1B*). In contrast to these decreases in energy expenditure, CR *increased* total and daytime physical activity in both sexes, with CR females having higher activity than CR males during week 1 (*Figure 3—figure supplement 1C*). Together, these data show that CR decreases energy expenditure more in females than in males, despite increasing physical activity, and that factors beyond decreased lean mass contribute to this CR effect.

We next analysed the respiratory exchange ratio (RER) to assess how diet and sex influence carbohydrate and lipid oxidation, and if these effects differ as CR progresses. In AL mice, RER peaked during the night, when most food consumption and physical activity occurs; in contrast, RER for CR mice peaked from 12.00 to 17.00 in the daytime, following provision of the daily ration of CR diet (*Figure 3D–E*; *Figure 3—figure supplement 2A*). The effects of CR on average daytime, night-time or total RER did not differ between the sexes (*Figure 3E*, *Figure 3—figure supplement 2A*); however, the dynamic changes in RER during these periods, and the strong influence of feeding and fasting on RER in the CR mice, make such average RER measurements difficult to interpret. Thus, we further investigated if RER in the postprandial and/or fasted states differs between CR males and females. Postprandial RER, calculated as the average RER from 12.00 to 17.00, exceeded 1 in both sexes (*Figure 3D and F*), indicating that CR mice use dietary carbohydrates for FA synthesis (*Bruss et al., 2010*). Notably, this postprandial RER was greater in females than in males, particularly during week 1 (*Figure 3F*). Fasting RER, calculated as the average from 04.00 to 09.00, was below 0.8 for both sexes, indicating preferential oxidation of lipids rather than carbohydrates during this fasted state (*Figure 3—figure supplement 2B*). Fasting RER was lower during week 1 than during week 3 of CR but, unlike for postprandial RER, no sex differences were observed (*Figure 3—figure supplement 2B*). This suggests that relative lipid oxidation is greater in the first week of CR and is similar in males

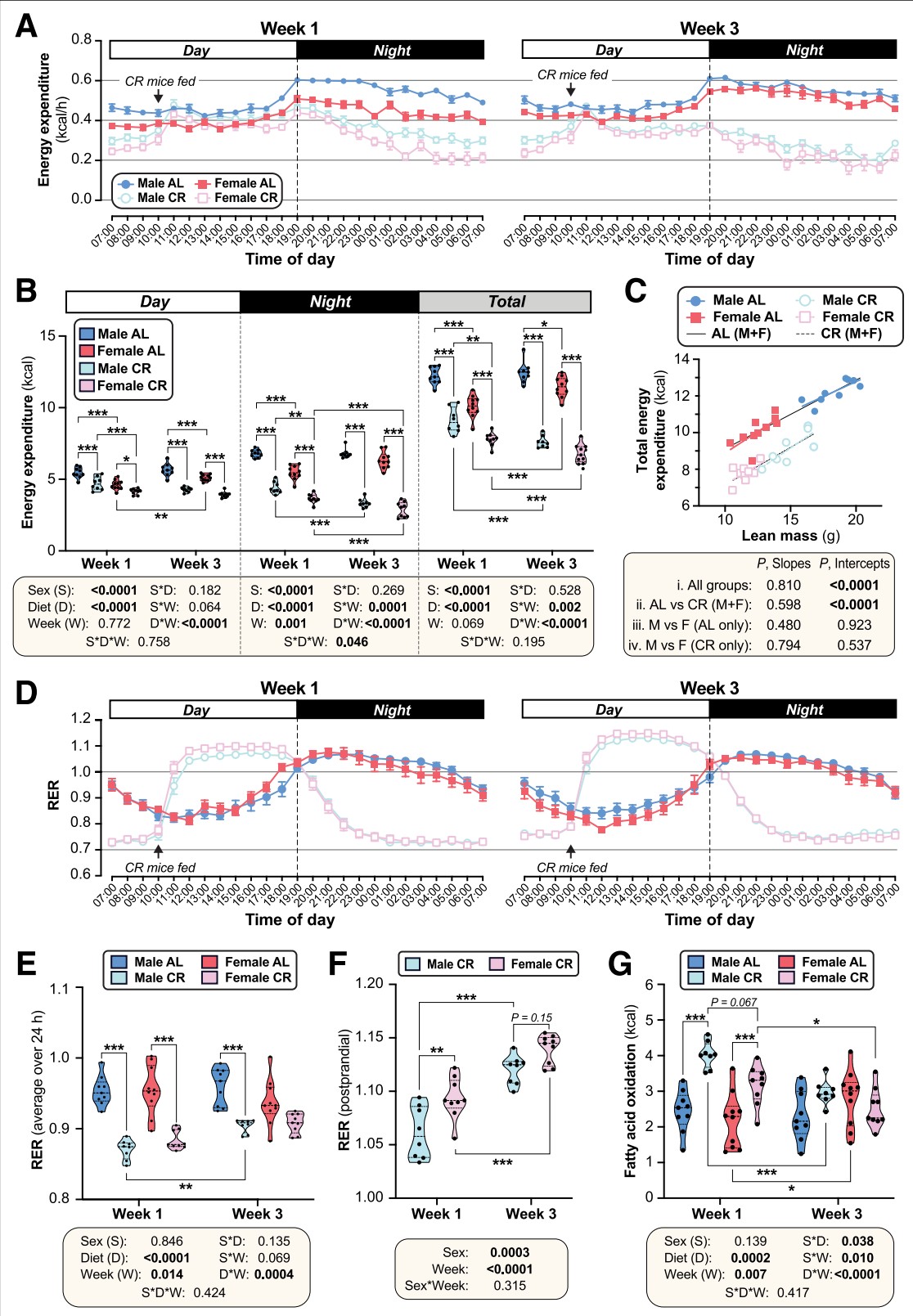

**Figure 3.** CR decreases energy expenditure and stimulates postprandial lipogenesis more in female than in male mice. Male and female mice were fed AL or CR diets, as described for *Figure 2*. In week 1 and week 3 after beginning AL or CR diets, mice were housed for 4 days in Promethion CORE System cages for indirect calorimetry. Energy expenditure (**A–C**) and respiratory exchange ratio (RER; **D–F**) was recorded every minute throughout the 4 days. (**A,D**) Average energy expenditure (**A**) (kcal) or RER (**D**) per hour over the 24 hr light (Day) and dark (Night) periods, based on the average for

*Figure 3 continued on next page*

*Figure 3 continued*

days 2–4 of Promethion housing, for Week 1 (left) and Week 3 (right). (**B**) Overall energy expenditure (kcal) during the day, night, or day +night (Total) for Week 1 and Week 3. (**C**) Linear regression of lean mass vs total energy expenditure (kcal/24 h) during Week 1. (**E**) Average total RER (day +night) in Weeks 1 and 3. (**F**) Average RER during the postprandial period, from 12.00 to 17.00, for CR mice. (**G**) Absolute FA oxidation was determined based on energy expenditure and RER as described (*Bruss et al., 2010*). Data are from 10 (*female AL*), 9 (*female CR, male AL*) or 8 (*male CR*) mice per group. In (**A**) and (**D**), data are shown as mean ± SEM. In (**B**) and (**E–G**), data are shown as violin plots overlaid with individual data points; within each time period (day, night, or total), significant effects of sex, diet, week, and interactions thereof, were determined by three-way (**B,E,G**) or two-way ANOVA (**F**), with p values shown beneath each graph. Statistically significant differences between comparable groups were further assessed by Šídák's (**B,E,G**) or Tukey's (**F**) multiple comparisons tests and are indicated by * (p<0.05), ** (p<0.01), or *** (p<0.001). For linear regression in (**C**), ANCOVA was used to test if the relationship between lean mass and total energy expenditure differs significantly across all of the individual diet-sex groups (*i. All mice*); between AL and CR mice, irrespective of sex (*ii. AL vs CR (M+F)*); and between males and females fed AL diet (*iii*) or CR diet (*iv*) only. ANCOVA p values for differences in slope and intercept are reported beneath the graph. See also *Figure 3—figure supplements 1–2*.

The online version of this article includes the following source data and figure supplement(s) for figure 3:

**Source data 1.** CR decreases energy expenditure and stimulates postprandial lipogenesis more in female than in male mice.

**Figure supplement 1.** Effects of CR on energy expenditure and physical activity.

**Figure supplement 1—source data 1.** Effects of CR on energy expenditure and physical activity.

**Figure supplement 2.** Effects of CR on Respiratory exchange ratio and FA oxidation.

**Figure supplement 2—source data 1.** Effects of CR on Respiratory exchange ratio and FA oxidation.

and females. However, given that females have lower absolute energy expenditure, we hypothesised that they would also have lower absolute FA oxidation (*Bruss et al., 2010*). As shown in *Figure 3G* and *Figure 3—figure supplement 2C*, CR increased absolute FA oxidation in both sexes during week 1, but not during week 3. Across both timepoints there was a significant sex-diet interaction, with CR increasing FA oxidation more in males than females (*Figure 3G*). Together, these data show that females have greater postprandial FA synthesis and lower absolute FA oxidation than males, particularly during the first week of CR. This highlights further mechanisms through which females maintain fat mass during CR.

## Females resist CR-induced improvements in glucose homeostasis

We next investigated if other metabolic effects of CR also differ between the sexes. We found that CR decreased blood glucose to a greater extent in males than in females (*Figure 4A*). Consistent with this, oral glucose tolerance tests (OGTT) revealed that CR improved glucose tolerance in both sexes, but this effect was greater in males (*Figure 4B*). These diet and sex effects were reflected by the OGTT total area under the curve (tAUC), calculated against 0 mM blood glucose; however, they were not apparent for the incremental AUC (iAUC), calculated against fasting blood glucose for each mouse (*Figure 4C*). This suggests that differences in fasting glucose are the main driver of CR-induced improvements in glucose tolerance and the sex differences therein.

To further assess the basis for these effects on glucose tolerance, we analysed plasma insulin during the OGTT. CR decreased plasma insulin in males but not in females, largely because AL females were already relatively hypoinsulinaemic (*Figure 4D*). Based on these blood glucose and insulin concentrations we calculated the homeostasis model assessment for insulin resistance (HOMA-IR) (*Matthews et al., 1985*) and the Matsuda Index of insulin sensitivity (*Matsuda and DeFronzo, 1999*). Here, a lower HOMA-IR score or a higher Matsuda index indicate increased insulin sensitivity. Across both sexes CR significantly decreased HOMA-IR and increased the Matsuda Index; however, within each sex the CR effect was significant for males only, indicating that females resist CR-induced improvements in insulin sensitivity (*Figure 4E*).

## Time of CR feeding does not alter the sex differences in CR's metabolic effects

Time of feeding can influence the metabolic effects of CR (*Speakman and Mitchell, 2011*; *Zhang et al., 2019*). Therefore, to test if our feeding regime influenced the observed sex differences, we studied a cohort of mice that received their daily ration of CR diet in the evening instead of the morning (*Figure 2—figure supplement 4*, *Figure 4—figure supplement 1*). Consistently, we found that the sex-dependent effect of CR on body mass, fat mass, lean mass and WAT mass was conserved

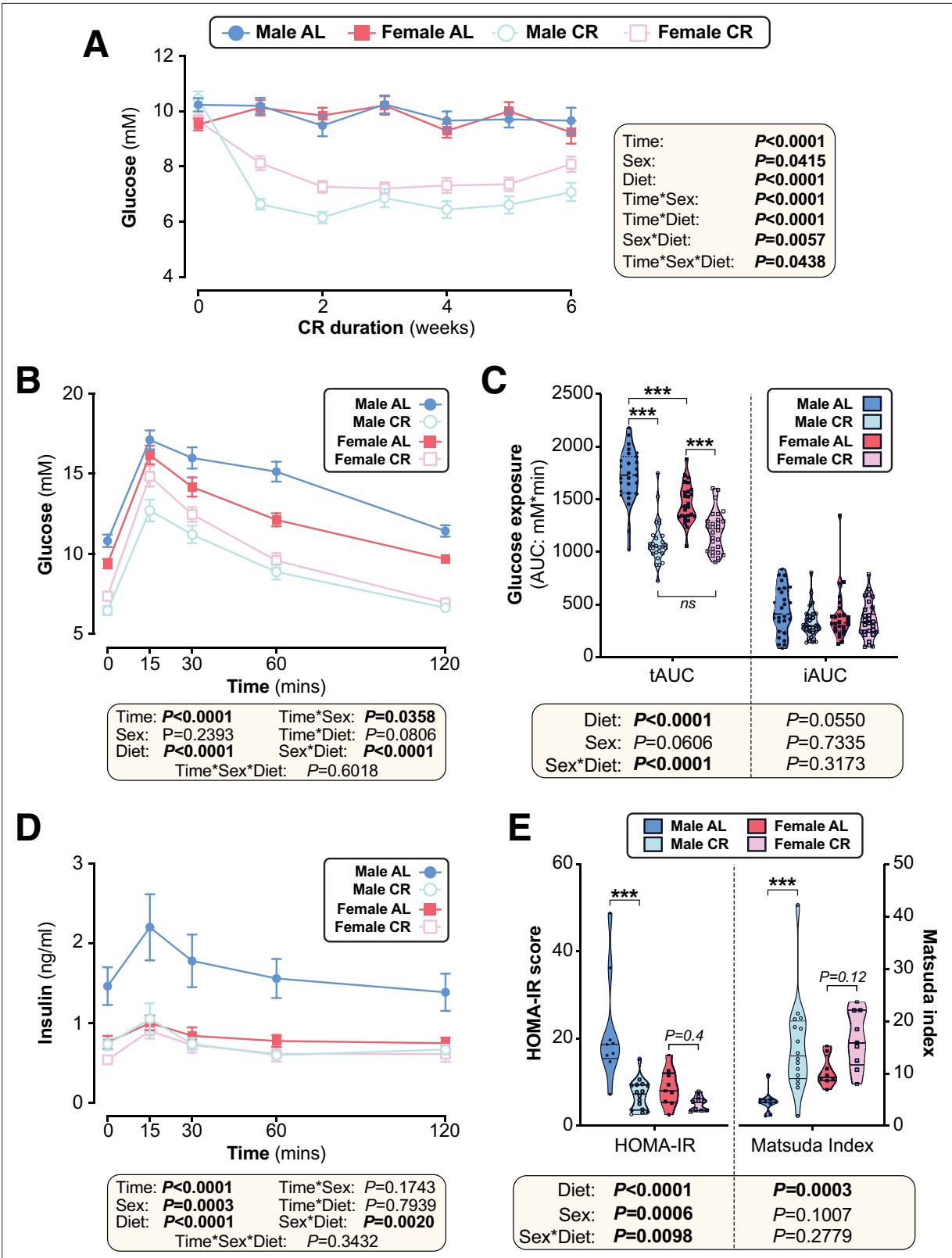

**Figure 4.** The effects of CR on glucose homeostasis differ between young male and female mice. Male and female C57BL6/NCrl mice were fed AL or CR diet from 9 to 15 weeks of age, as described for *Figure 2*. (**A**) Random-fed blood glucose was recorded each week. (**B–D**) At 13 weeks of age, mice underwent an oral glucose tolerance test (OGTT). (**B**) Blood glucose readings during the OGTT. (**C**) Area under the curve (AUC) during the OGTT was determined relative to 0 mmol/L (total AUC: tAUC) and relative to baseline (incremental AUC: iAUC). (**D**) Glucose-stimulated insulin secretion in mice

*Figure 4 continued on next page*

*Figure 4 continued*

during OGTT was assessed using an insulin ELISA. (**E**) HOMA-IR and Matsuda indices of mice calculated from glucose and insulin concentrations during the OGTT. Data in (**A**), (**B**), and (**D**) are presented as mean ± SEM. Data in (**C**) and (**E**) are presented as violin plots overlaid with individual data points. For each group and timepoint, the following numbers of mice were used: (**A**): *male AL*, n=42 (Wk 0), 36 (Wk2, Wk 2), 34 (Wk 3), 29 (Wk 4), 26 (Wk 5), or 22 (Wk 6); *female AL*, n=43 (Wk 0), 28 (Wk 1), 35 (Wk 2), 31 (Wk 3), 27 (Wk 4), 26 (Wk 5), or 21 (Wk 6); *male CR*, n=44 (Wk 0), 40 (Wk 1), 38 (Wk 2), 35 (Wk 3), 31 (Wk 4), 29 (Wk 5), or 26 (Wk 6); *female CR*, n=51 (Wk 0), 44 (Wk 1, Wk 2), 41 (Wk 4), 35 (Wk 4), 33 (Wk 5), or 27 (Wk 6). (**B**): *male AL*, n=27 ($T_0$, $T_{15}$, $T_{30}$, $T_{60}$, $T_{120}$); *female AL*, n=26 ($T_{15}$, $T_{30}$, $T_{120}$), or 25 ($T_0$, $T_{60}$); *male CR*, n=27 ($T_0$, $T_{15}$, $T_{30}$, $T_{60}$, $T_{120}$); *female CR*, n=28 ($T_0$, $T_{15}$, $T_{30}$, $T_{60}$, $T_{120}$). (**C**): *male AL*, n=27; *female AL*, n=26; *male CR*, n=27; *female CR*, n=28. (**D**): *male AL*, n=9 ($T_0$, $T_{15}$, $T_{30}$, $T_{60}$, $T_{120}$); *female AL*, n=11 ($T_0$, $T_{15}$), 10 ($T_{30}$, $T_{60}$), or 8 ($T_{120}$); *male CR*, n=17 ($T_0$), 16 ($T_{30}$), 15 ($T_{60}$), 13 ($T_{15}$), or 11 ($T_{120}$); *female CR*, n=11 ($T_{15}$, $T_{30}$, $T_{60}$), or 10 ($T_0$, $T_{120}$). (**E**): *male AL*, n=9; *female AL*, n=9; *male CR*, n=16; *female CR*, n=9. In (**A**), (**B**), and (**D**), significant effects of diet, sex and/or time, and interactions thereof, were determined using a mixed-effects model. In (**C**) and (**E**), significant effects of sex, diet, and sex-diet interaction for tAUC, iAUC, HOMA-IR, and Matsuda Index were assessed using two-way ANOVA with Tukey's multiple comparisons test. Overall p values for each variable, and their interactions, are shown with each graph. For (**C**) and (**E**), statistically significant differences between comparable groups are indicated by *** (p<0.001) or with p values shown. Source data are provided as a Source Data file. See also *Figure 4—figure supplement 1*.

The online version of this article includes the following source data and figure supplement(s) for figure 4:

**Source data 1.** The effects of CR on glucose homeostasis differ between young male and female mice.

**Figure supplement 1.** Sex differences in the effects of CR on glucose homeostasis persist when CR mice are fed in the evening.

**Figure supplement 1—source data 1.** Sex differences in the effects of CR on glucose homeostasis persist when CR mice are fed in the evening.

---

(*Figure 2—figure supplement 4A–H*). Similarly, an OGTT revealed that the diet and sex effects on glucose homeostasis persisted (*Figure 4—figure supplement 1A–B*).

## Sex differences in glucose homeostasis are not explained by differential glucose uptake

The basis for the sex differences in CR-improved glucose homeostasis was next investigated. We first used positron emission tomography-computed tomography (PET/CT) with $^{18}$F-fluorodeoxyglucose ($^{18}$F-FDG) to assess glucose uptake in vivo (*Figure 5*, *Figure 5—figure supplement 1*). We focused on tissues with high basal and/or insulin-stimulated glucose uptake, including the heart, brain, skeletal muscle, BAT, WAT, bone, and the bone marrow (BM; *Bartelt et al., 2017*); the latter was also included because CR increases bone marrow adipose tissue (BMAT), another site of high basal glucose uptake (*Suchacki et al., 2020*).

Consistent with previous observations (*Suchacki et al., 2020*), we found that glucose uptake was highest in BAT, followed by the brain, bones, BM, the heart and skeletal muscle (*Figure 5A–B*, *Figure 5—figure supplement 1A*). However, CR did not stimulate glucose uptake in any of these tissues and decreased tibial glucose uptake (*Figure 5A*), including in the BMAT-enriched distal tibia (*Figure 5—figure supplement 1A*). Gamma counting further revealed no significant effect of CR in the gastrocnemius muscle, arm bones, liver, pancreas, spleen, or thymus (*Figure 5C*, *Figure 5—figure supplement 1B*); however, across both sexes, CR tended to increase glucose uptake in iWAT and gWAT and this effect was significant when assessed across all WAT depots (*Figure 5C*). CR also significantly increased glucose uptake in the kidneys (*Figure 5C*). Several sex differences were further detected, independent of diet. Thus, females had lower glucose uptake into gastrocnemius muscle and higher glucose uptake into BAT, the heart, gWAT, the humerus, the forearm and the kidneys (*Figure 5A–C*). Notably, however, sex did not significantly alter the effect of CR in any of the tissues assessed. Thus, the sexually dimorphic effects of CR on glucose homeostasis are not a result of CR increasing glucose disposal to a greater extent in males than in females.

## CR exerts sexually dimorphic effects on hepatic sphingolipid content

The lack of CR-driven differences in peripheral uptake suggested that sex differences in glucose homeostasis relate instead to effects on hepatic insulin sensitivity and glucose production. Therefore, we next tested whether altered lipid storage, implicated in sex-dependent differences in hepatic glucose production (*Della Torre et al., 2018*), might contribute to the sexually dimorphic metabolic effects of CR. Because gross hepatic triglyceride content was not affected by diet or sex (*Figure 6A*), we undertook a more-detailed molecular investigation of lipid species that may underpin altered hepatic glucose regulation, focussing on sphingolipids (*Petersen and Shulman, 2017*; *Chaurasia et al., 2019*). Targeted lipidomics revealed that total ceramide content was significantly lower in

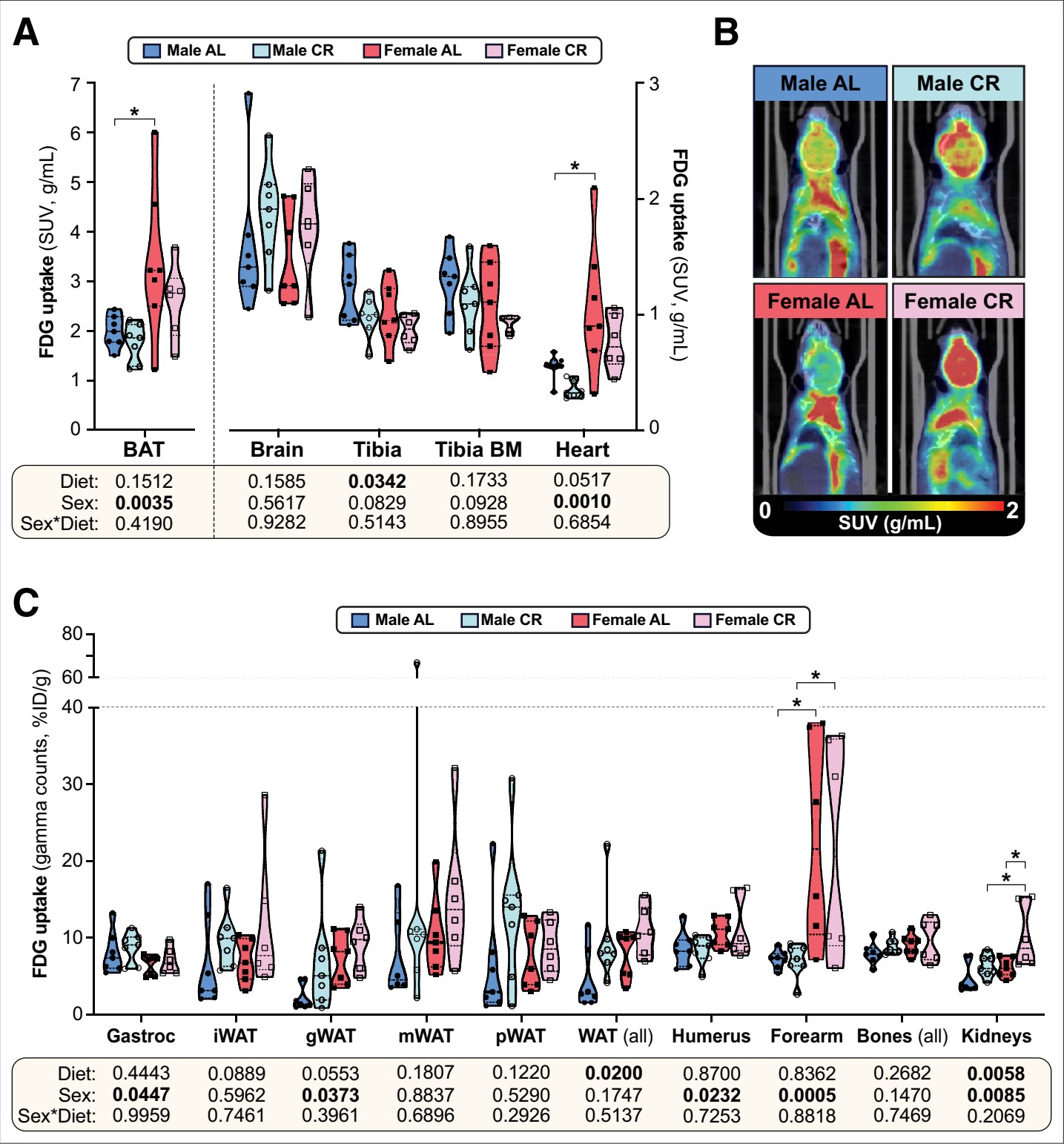

**Figure 5.** The effects of CR on glucose uptake do not differ between male and female mice. Male and female C57BL6/NCrl mice were fed AL or CR diet from 9 to 15 weeks of age, as described for *Figure 2*. At 15 weeks of age, glucose uptake was assessed by PET/CT with ¹⁸F-fluorodeoxyglucose (¹⁸F-FDG). (**A–C**) ¹⁸F-FDG uptake in the indicated tissues was determined as SUVs from PET/CT scans (**A**) or as % injected dose per gram of tissue (%ID/g) from gamma counting of dissected whole tissues (**C**). (**B**) PET/CT images confirm that ¹⁸F-FDG uptake in interscapular BAT is greater in females than in males, regardless of diet. Data are shown as violin plots (**A,C**) or representative images (**B**) of 7 (*male AL, female AL, male CR*) or 6 (*female CR*) mice per group. For each tissue in (**A**) and (**C**), significant effects of sex, diet, and sex-diet interaction were assessed by two-way ANOVA; overall p values for

*Figure 5 continued on next page*

*Figure 5 continued*

each variable, and their interactions, are shown beneath each graph. Significant differences between each group, as determined using Tukey's multiple comparisons test, are indicated by * (p<0.05). Source data are provided as a Source Data file. See also *Figure 5—figure supplement 1*.

The online version of this article includes the following source data and figure supplement(s) for figure 5:

**Source data 1.** The effects of CR on glucose uptake do not differ between male and female mice.

**Figure supplement 1.** The effects of CR on glucose uptake in peripheral tissues.

**Figure supplement 1—source data 1.** The effects of CR on glucose uptake in peripheral tissues.

females than in males but was not influenced by CR (*Figure 6B*); however, individual ceramide species were modulated by CR in a sex-dependent manner. Thus, 22:0 and 22:1 ceramides were lower in females than in males and were decreased by CR, with this diet effect being stronger in males (*Figure 6B*, *Figure 6—figure supplement 1A*). Conversely, 23:0 ceramides were greater in AL females than in AL males and were increased by CR in males only (*Figure 6B*); a similar pattern occurred for 14:0, 16:0, 18:0, 18:1, 20:1, 26:0, and 26:1 ceramides (*Figure 6—figure supplement 1A*).

Ceramides are generated from dihydroceramides (DHCs) and an elevated hepatic ceramide:DHC ratio promotes metabolic dysfunction and insulin resistance (*Chaurasia et al., 2019*; *Apostolopoulou et al., 2020*). Thus, we further analysed DHCs and the ceramide:DHC ratio as potential mediators of CR's sexually dimorphic metabolic effects. As for total ceramides, total DHCs were lower in females than in males; however, unlike for total ceramides, CR increased total DHCs in males only (*Figure 6C*). Differential effects of sex and/or diet also occurred for the specific DHC species. A common pattern was for CR to increase DHC concentrations in males only, such that these were lowest in AL males but similar in CR males, CR females and AL females; this pattern occurred for 16:0, 18:0, 18:1, 20:0, 23:0, 24:0, and 24:1 DHC species (*Figure 6C*, *Figure 6—figure supplement 1B*). CR increased 14:0 DHCs in both sexes, albeit more strongly in males, and decreased 22:0 DHCs across both sexes (*Figure 6C*, *Figure 6—figure supplement 1B*). The overall ceramide:DHC ratio was decreased by CR in males but not in females (*Figure 6C*) and this also occurred for many of the individual species, including 16:0, 18:0, 20:0, 22:0, 22:1, 23:0, 24:0 and 24:1 (*Figure 6—figure supplement 1B*); however, unlike for the total ceramide:DHC ratio, the ratio for each of these species was highest in the AL males and significantly lower in females, such that CR males had a similar ratio to females on either diet (*Figure 6—figure supplement 1B*). Consistent with this, principal component analysis of these ceramide data identified AL males as the most distinct group, with CR males being more similar to females on either diet (*Figure 6D*).

Together, these data indicate that CR suppresses ceramide synthesis to a greater extent in males than in females, highlighting a potential mechanism for CR's sexually dimorphic effects on glucose homeostasis.

## CR exerts sexually dimorphic effects on hepatic gene expression and ketogenesis

To further investigate how hepatic function contributes to the sex differences in glucose homeostasis, we used RNA-seq to characterise the livers of AL and CR mice. Principal component analysis of this RNA-seq data identified AL males as the most transcriptionally distinct group, while CR males were more similar to AL or CR females (*Figure 7A*). Volcano plots further revealed the substantial effects of sex and diet on hepatic gene expression (*Figure 7B*, *Figure 7—figure supplement 1A–B*): hundreds of genes were differentially expressed between males and females within each diet group (*Figure 7B*, *Figure 7—figure supplement 1B*) and between AL and CR mice, both within and across the two sexes (*Figure 7—figure supplement 1A–B*). To assess the metabolic implications of these transcriptional differences, we first analysed the expression of transcripts that encode proteins associated with gluconeogenesis, glycolysis, or the TCA cycle. Few significant changes were detected for individual transcripts (*Figure 7B*, *Figure 7—figure supplement 1A*), and therefore we used gene set enrichment analysis (GSEA) to assess higher-level transcriptional changes associated with these and other relevant metabolic pathways (*Figure 7C*). Gluconeogenesis-related genes were significantly increased with CR in each sex, but, against our expectations, they were more highly expressed in males than in females during CR (*Figure 7C*). The greatest difference between CR males and CR females was for genes related to oxidative phosphorylation and the TCA cycle, which increased with CR in males but not in

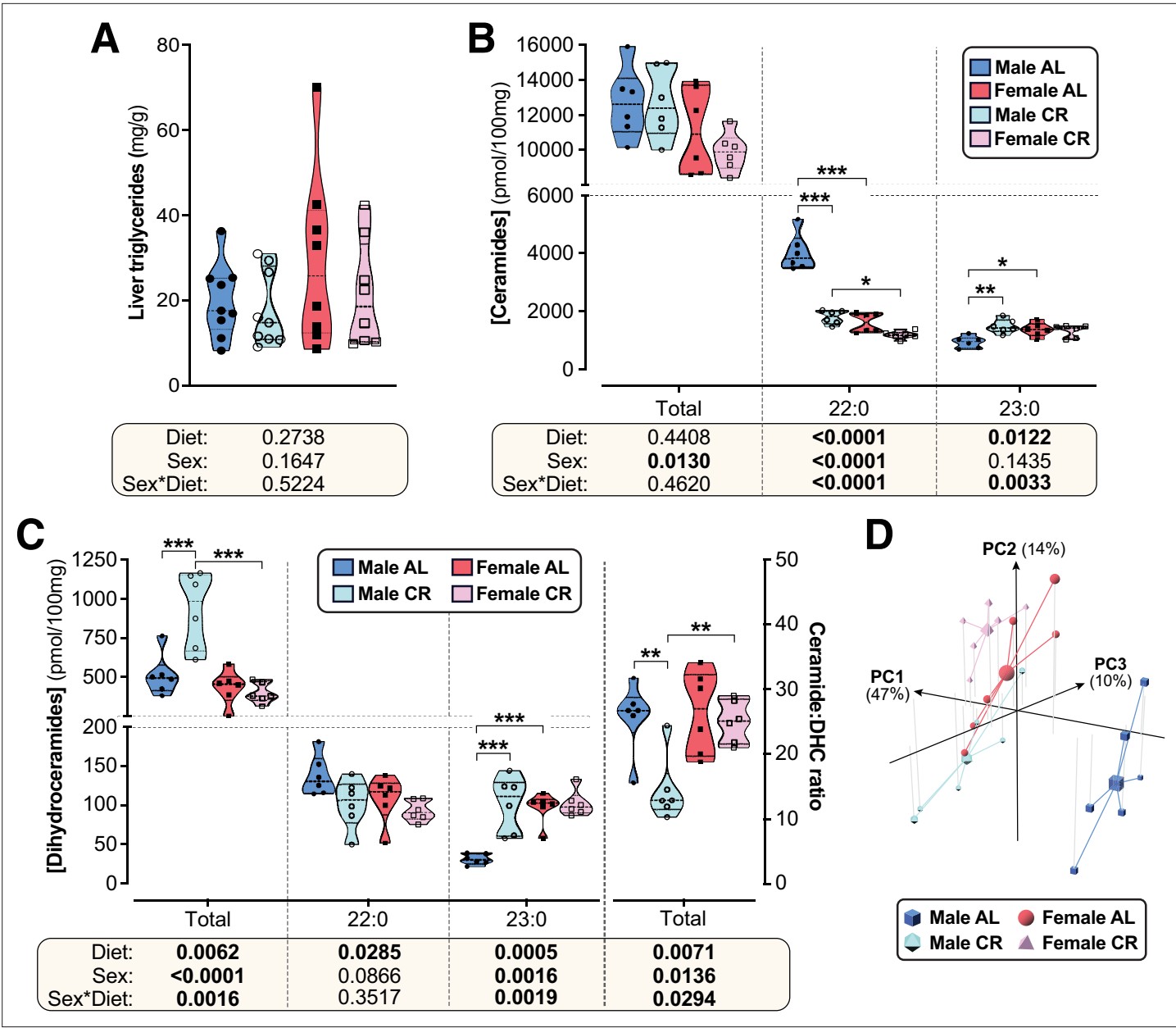

**Figure 6.** CR exerts sexually dimorphic effects on hepatic sphingolipid content. Male and female C57BL6/NCrl mice were fed AL or CR diet from 9 to 15 weeks of age, as described for *Figure 2*. Livers were sampled at necropsy (15 weeks) and used to assess triglyceride, ceramide, and dihydroceramide. (**A**) Hepatic triglyceride content. (**B,C**) LC-MS analysis of total, 22:0 and 23:0 ceramides (**B**), dihydroceramides (**C**, left side) and the ceramide:DHC ratio (**C**, right side). (**D**) Principal component analysis of ceramide and dihydroceramide content based on data in (**B–C**) and *Figure 6—figure supplement 1*. PC1, PC2, and PC3 account for 47.4%, 14.1%, and 10.5% of the variance, respectively. Data in (**A**), (**B**), and (**C**) are presented as truncated violin plots overlaid with individual data points. Data represent the following numbers of mice per group. (**A**): *male AL*, n=9; *female AL*, n=8; *male CR*, n=9; *female CR*, n=8. (**B–D**): 6 mice per group. For (**A**), (**B**), and (**C**), significant effects of sex, diet, and sex-diet interaction were assessed using two-way ANOVA; overall p values for each variable, and their interactions, are shown beneath each graph. Significant differences between comparable groups were assess using Tukey's multiple comparison test and are indicated by * (p<0.05), ** (p<0.01), or *** (p<0.001). Source data are provided as a Source Data file. See also *Figure 6—figure supplement 1*.

The online version of this article includes the following source data and figure supplement(s) for figure 6:

**Source data 1.** CR exerts sexually dimorphic effects on hepatic sphingolipid content.

**Figure supplement 1.** CR exerts sexually dimorphic effects on hepatic sphingolipid content.

**Figure supplement 1—source data 1.** CR exerts sexually dimorphic effects on hepatic sphingolipid content.

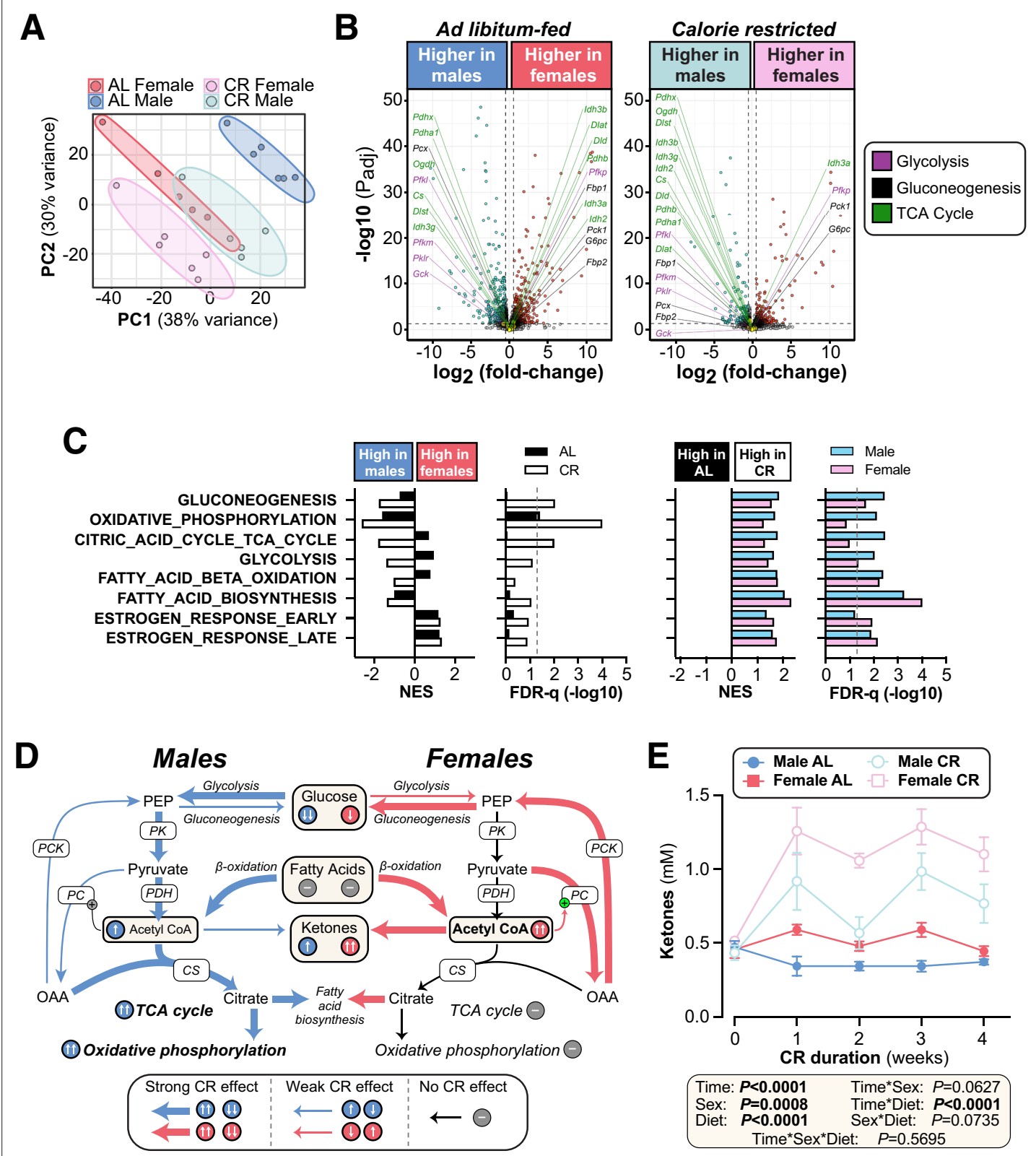

**Figure 7.** CR exerts sexually dimorphic effects on hepatic gene expression and blood ketone levels. Male and female C57BL6/NCrl mice were fed AL or CR diet from 9 to 13 weeks of age, as described for *Figure 2*. Blood ketone concentrations were assessed weekly. Livers were sampled at necropsy (13 weeks) and analysed by RNA-seq. (**A**) Principal component analysis of all samples. PC1 and PC2 account for 38% and 30% of the variance, respectively. (**B**) Volcano plots showing differentially expressed genes between AL males and AL females (left) or between CR males and CR females

*Figure 7 continued on next page*

*Figure 7 continued*

(right). Transcripts encoding key proteins in gluconeogenesis, glycolysis, or the TCA cycle are shown on each graph. The names of genes with positive fold-change values are shown on the right side and those with negative fold-change values are shown on the left side of the volcano plots; none of these genes had absolute fold-change >1.5 and $P$adj <0.05. (**C**) GSEA results for gene sets indicative of relevant pathways. Significant sex differences (within AL or CR diets) are shown on the left; significant diet differences (within males or females) are shown on the right; NES = normalised enrichment score. (**D**) Schema showing the proposed sex differences in liver metabolism during CR, based on the RNA-seq data. PK, pyruvate kinase; PDH, pyruvate dehydrogenase; CS, citrate synthase; PC, pyruvate carboxylase; PCK, phosphoenolpyruvate carboxykinase. (**E**) Blood ketone concentrations. Data represent the following numbers of mice per group: (**A–C**) – *male AL*, n=6; *female AL*, n=5; *male CR*, n=5; *female CR*, n=6. (**E**) – *male AL*, n=7; *female AL*, n=9; *male CR*, n=6; *female CR*, n=7. For (**A**) and (**B**), R script are provided as a Source Data file. For (**C**), gene set collection files are prepared with the 8 gene sets shown for GSEA. For (**E**), significant effects of sex, diet, and sex-diet interaction were assessed using three-way ANOVA; overall p values for each variable, and their interactions, are shown beneath each graph. Source data are provided as a Source Data file. See also *Figure 7—figure supplement 1*.

The online version of this article includes the following source data and figure supplement(s) for figure 7:

**Source data 1.** CR exerts sexually dimorphic effects on hepatic gene expression and blood ketone levels.

**Figure supplement 1.** Effects of sex and diet on the hepatic transcriptome.

**Figure supplement 1—source data 1.** Effects of sex and diet on the hepatic transcriptome.

females (*Figure 7C*); this was also apparent from the distribution of individual TCA-related genes in the volcano plots (*Figure 7B*, *Figure 7—figure supplement 1A*). Glycolysis-related genes were also increased by CR, more so in males, whereas transcripts relevant to FA β-oxidation and FA biosynthesis were increased by CR in both sexes (*Figure 7C*, *Figure 7—figure supplement 1C–D*). These gene expression profiles support the following model (*Figure 7D*): during CR, both sexes increase FA β-oxidation to generate acetyl-CoA and this is supported by increased FA biosynthesis. Males, more than females, also increase glycolysis for acetyl-CoA production; however, males utilise the acetyl-CoA to support increased TCA cycle activity and oxidative phosphorylation, pathways that are not increased by CR in females. If so, females should have greater hepatic acetyl-CoA levels than males during CR. To test this, we measured plasma ketone concentrations, which are a biomarker for hepatic acetyl-CoA (*Perry et al., 2017*). Consistent with our model, plasma ketones were significantly higher in females than males, particularly during CR (*Figure 7E*). Hepatic acetyl-CoA stimulates gluconeogenesis by activating pyruvate carboxylase; thus, despite transcriptional profiles suggesting increased gluconeogenesis in males (*Figure 7C*), females' higher levels of acetyl-CoA would be expected to cause greater stimulation of gluconeogenesis, thereby explaining why, compared to males, females are better at maintaining blood glucose during CR (*Figure 7D*).

## Sex differences in CR's metabolic effects are absent when CR is initiated in aged mice

Another intriguing finding from our RNA-seq data is that CR significantly stimulates oestrogen response pathways (*Figure 7C*). Thus, one possibility is that young females resist many of CR's metabolic effects because their endogenous oestrogens cause them to remain relatively metabolically healthy, even on an AL diet (*Forney et al., 2020*; *Chaix et al., 2021*). To begin testing if endogenous oestrogens influence the CR response, we next investigated the effects of CR initiated in 18-month-old mice, when females are anoestrous (*Gee et al., 1983*). Unlike in young mice, CR decreased body mass and fat mass to a similar extent in aged males and females, while the decreases in lean mass were greater in aged females (*Figure 8A–C*). Consistent with this, CR's effects on the masses of individual WAT depots did not differ between the sexes (*Figure 8D*, *Figure 8—figure supplement 1A*). Among non-adipose tissues CR significantly decreased pancreas mass only in aged females, but there were no sex differences in the effects of CR on the masses of other tissues (*Figure 8—figure supplement 1B–C*). Thus, unlike in young mice, aged males and females show a similar effect of CR on overall adiposity and this is driven predominantly by loss of WAT.

We then investigated CR's effects on glucose homeostasis and insulin sensitivity. As shown in *Figure 9A*, in aged mice the ability of CR to decrease fasting blood glucose was not significantly influenced by sex. CR improved glucose tolerance to a similar extent in aged males and females, with tAUC but not iAUC being significantly decreased with CR (*Figure 9B–C*). The latter indicates that, as for young mice, CR improves glucose tolerance in aged mice primarily by decreasing fasting glucose. Despite no sex differences in glucose tolerance, both sex and diet decreased glucose-stimulated

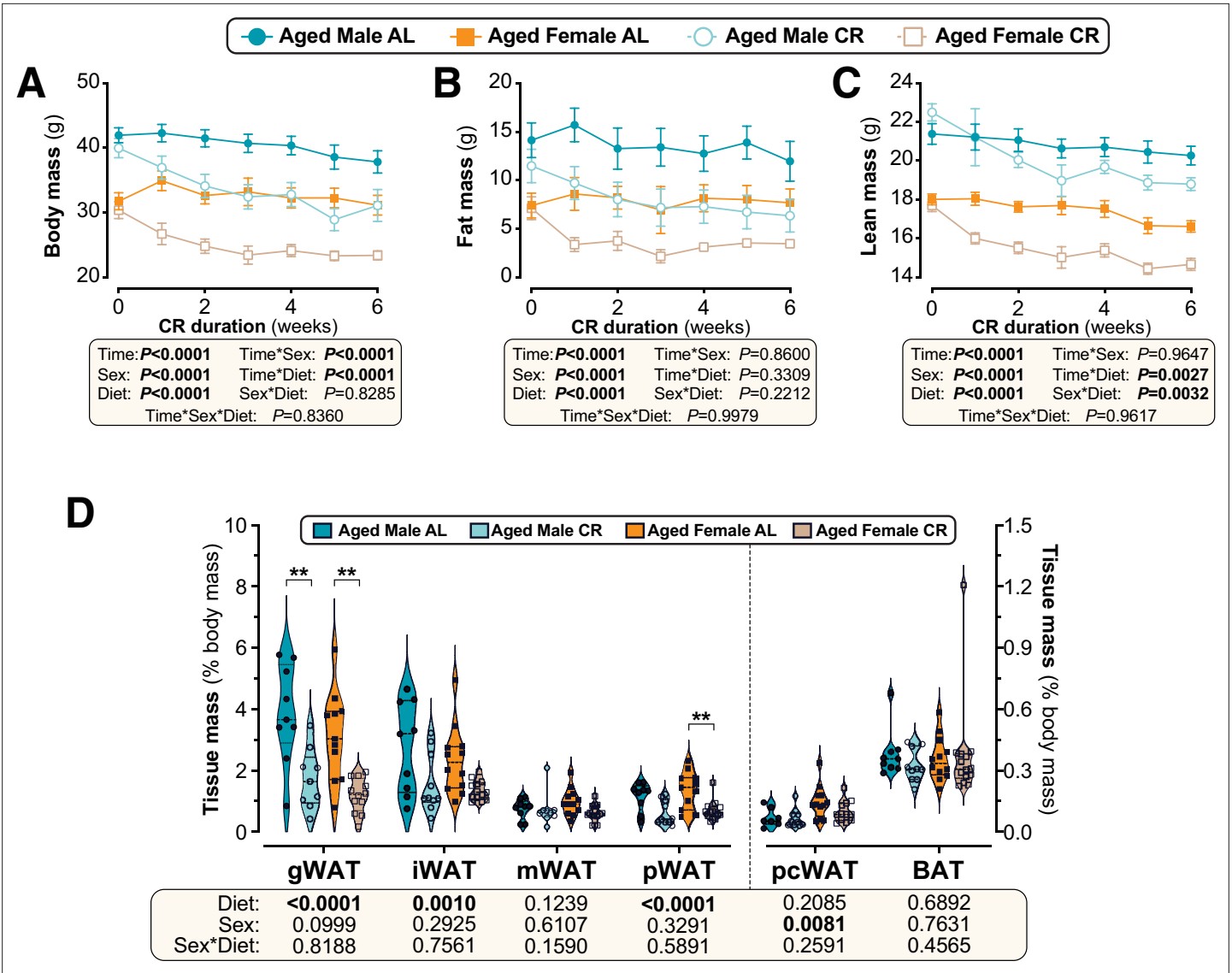

**Figure 8.** Sex differences in CR-induced weight loss and fat loss are absent when CR is initiated in aged mice. Male and female C57BL6/NCrl mice were single housed and fed AL or a CR diet (70% of daily AL intake) from 78 to 84 weeks of age. (A–F) Each week mice were weighed (A,B) and body composition was determined by TD-NMR (C–F). Body mass, fat mass and lean mass are shown as absolute masses (A,C,E) or fold-change relative to baseline (B,D,F). (G) Masses of gWAT, iWAT, mWAT, pWAT, pcWAT, and BAT were recorded at necropsy and are shown as % body mass. Data in (A–F) are shown as mean ± SEM. Data in (G) are shown as violin plots overlaid with individual data points. For each group and timepoint, the following numbers of mice were used: (A,B): *male AL,* n=9 (Wk 0, Wk 2, Wk3, Wk 4), 8 (Wk 1), or 6 (Wk 5, Wk 6); *female AL,* n=13 (Wk 0, Wk 2), 12 (Wk 4, Wk 5), 11 (Wk 6), 9 (Wk 1), or 8 (Wk 3); *male CR,* n=10 (Wk 0, Wk 2–4), 9 (Wk 1), 7 (Wk 5), or 6 (Wk 6); *female CR,* n=14 (Wk 0, Wk 4, Wk 5), 13 (Wk 2, Wk 6), 10 (Wk 3), or 9 (Wk 1). (C–F): *male AL,* n=9 (Wk 0, Wk 2), 8 (Wk 3, Wk 4, Wk 6), or 7 (Wk 1, Wk 5); *female AL,* n=13 (Wk 0, Wk 2), 9 (Wk 4, Wk 6), 8 (Wk 5), 6 (Wk 1), or 5 (Wk 3); *male CR,* n=10 (Wk 0, Wk 2), 9 (Wk3-6), or 8 (Wk 1); *female CR,* n=14 (Wk 1, Wk 2), 11 (Wk 4–6), 7 (Wk 3), or 6 (Wk 1). (G): *male AL,* n=9; *female AL,* n=12 (iWAT, mWAT) or 11 (gWAT, pWAT); *male CR,* n=10 (iWAT, pWAT), or 9 (gWAT, mWAT); *female CR,* n=14 (mWAT, pWAT), 13 (iWAT), or 12 (gWAT). For (A–F), significant effects of diet, sex and/or time, and interactions thereof, were determined by three-way ANOVA or a mixed-effects model. For (G), significant effects of sex, diet, and sex-diet interaction were assessed using two-way ANOVA with Tukey's multiple comparisons test. Overall p values for each variable, and their interactions, are shown beneath each graph. For (G), significant differences between comparable groups are indicated by ** (p<0.01) or *** (p<0.001). Source data are provided as a Source Data file. See also *Figure 8—figure supplement 1*.

The online version of this article includes the following source data and figure supplement(s) for figure 8:

**Source data 1.** Sex differences in CR-induced weight loss and fat loss are absent when CR is initiated in aged mice.

**Figure supplement 1.** CR and sex effects on the masses of adipose and non-adipose tissues in aged mice.

**Figure supplement 1—source data 1.** CR and sex effects on the masses of adipose and non-adipose tissues in aged mice.

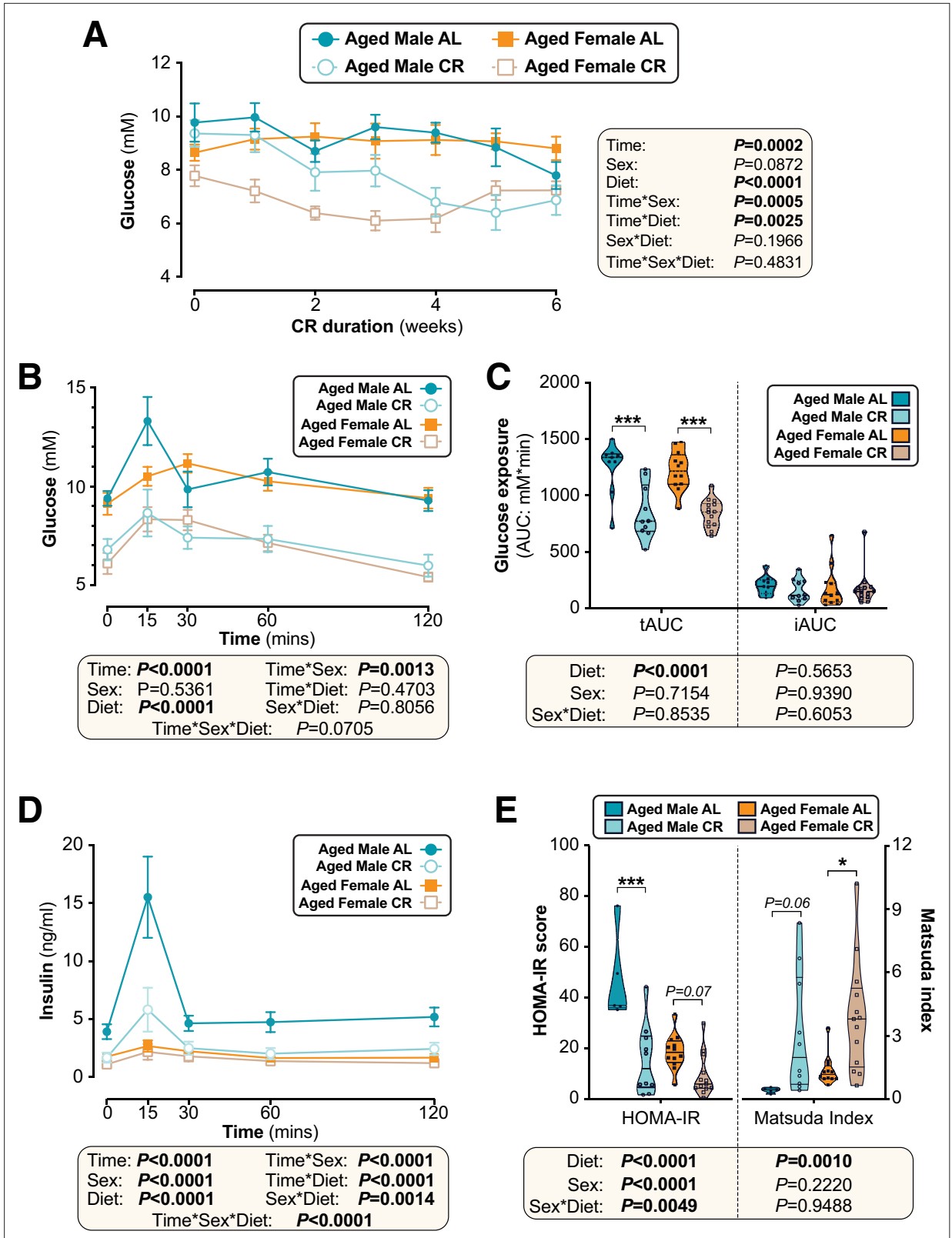

**Figure 9.** Sex differences in CR's effects on glucose homeostasis are largely absent when CR is initiated in aged mice. Male and female C57BL6/NCrl mice were fed AL or CR diet from 78 to 84 weeks of age, as described for *Figure 7*. (**A**) Random-fed blood glucose was recorded each week. (**B–D**) At 82 weeks of age mice underwent OGTTs. Blood glucose was recorded at each timepoint (**B**) and the tAUC and iAUC was determined (**C**). (**D**) Glucose-stimulated insulin secretion in mice during OGTT was assessed using an insulin ELISA. (**E**) HOMA-IR and Matsuda indices of mice calculated from

*Figure 9 continued on next page*

*Figure 9 continued*

glucose and insulin concentrations during the OGTT. Data in (**A**), (**B**), and (**D**) are presented as mean ± SEM. Data in (**C**) and (**E**) are presented as violin plots overlaid with individual data points. For each group and timepoint the following numbers of mice were used: (**A**): *male AL*, n=9 (Wk 0, Wk 2–4, Wk 6), 8 (Wk 1), or 5 (Wk 5); *female AL*, n=12 (Wk 0, Wk2, Wk4), 8 (Wk 1, Wk 3), or 11 (Wk 5, Wk 6); *male CR*, n=10 (Wk 0, Wk 2–4, Wk 6), 9 (Wk 1), or 7 (Wk 5); *female CR*, n=14 (Wk 0, Wk 2, Wk 4–6), 9 (Wk 1) or 10 (Wk 3). (**B–E**): *male AL*, n=9 (**B–C**) or 5 (**D–E**); *female AL*, n=12; *male CR*, n=10; *female CR*, n=14. In (**A**), (**B**), and (**D**), significant effects of diet, sex and/or time, and interactions thereof, were determined using a mixed-effects model or three-way ANOVA. In (**C**) and (**E**), significant effects of sex, diet, and sex-diet interaction for tAUC, iAUC, HOMA-IR, and Matsuda Index were assessed using two-way ANOVA with Tukey's multiple comparisons test. Overall p values for each variable, and their interactions, are shown with each graph. For (**C**) and (**E**), statistically significant differences between comparable groups are indicated by * (p<0.05) or *** (p<0.001), or with p values shown. Source data are provided as a Source Data file.

The online version of this article includes the following source data for figure 9:

**Source data 1.** Sex differences in CR's effects on glucose homeostasis are largely absent when CR is initiated in aged mice.

insulin secretion, with this being greatest in aged AL males and decreased by CR more so in males than in females (*Figure 9D*). Interestingly, a sex-diet effect was observed for HOMA-IR but not for the Matsuda Index (*Figure 9E*); this is similar to the effects observed in young mice (*Figure 4E*) and suggests that in aged mice there remain some sex differences in the relationship between plasma insulin and gluconeogenesis in the fasted state. However, overall, these studies demonstrate that sex differences in the metabolic effects of CR are highly age dependent, with CR being equally effective in aged males and females whereas young females resist these aspects of the CR response.

## In humans CR-induced fat loss is sex- and age-dependent

Finally, we investigated if the metabolic effects of CR in humans are also sex- and age-dependent. To do so, we retrospectively analysed data from 42 overweight and obese men and women who participated in a weight loss study involving a 4-week CR intervention. As in young mice, body mass and fat-free (lean) mass decreased more so in males than in females, with males also showing a trend for greater fold-decreases in total fat mass (*Figure 10A–F*). When adjusted for age (*Figure 10G*), fold-change in body mass differed significantly between males and females (*P*, Intercept = 0.025). This demonstrates that overall weight loss is greater in males than females, independent of age. In contrast, the decreases in fat mass and fat-free mass were sex- and age-dependent. Thus, fat loss in younger individuals was greater in males than females but, with age, fat loss increased in females but diminished in males (*Figure 10H*) (*P*, Slope = 0.007). Conversely, the loss of fat-free mass in younger individuals was greater in females than in males and this relationship reversed in older individuals (*Figure 10I*). As for fat mass, the relationship between age and loss of fat-free mass differed significantly between the sexes (*P*, Slope = 0.0004). In contrast, the sex differences in loss of fat mass or fat-free mass were unrelated to fat mass or BMI at baseline (*Figure 10—figure supplement 1*).

These age-sex interactions were particularly apparent when the cohort was divided into younger and older groups (<45 vs >45 years of age), selected based on the age when females' oestrogen levels begin to decline (*Diaz Brinton, 2012*). Total weight loss was greater in males than females regardless of age (*Figure 10J*), but only in the younger age group did males show greater fat loss than females (*Figure 10K*). Conversely, males' greater loss of fat-free mass occurred only in the older age group (*Figure 10L*). Notably, a similar pattern of age-dependent fat loss was apparent for CR in young vs aged mice (*Figure 10—figure supplement 2*). Thus, our data demonstrate that the ability of CR to decrease body mass, fat mass and lean mass is sex- and age-dependent in both mice and humans.

## Discussion

Herein we aimed to determine the extent and basis of sex differences in the metabolic effects of CR. We reveal that 96.6% (mouse) and 95.7% (human) of published CR research overlooks potential sex differences, with most mouse studies using males only whereas most human studies combine data from males and females. We show that, in young mice, CR decreases fat mass and improves glucose homeostasis to a greater extent in males than in females, whereas these sex differences are blunted or absent when CR is initiated in old mice. Young CR females likely resist weight loss and fat loss by having lower energy expenditure, lipolysis and FA oxidation, and greater postprandial lipogenesis, than young CR males, The effects of CR on hepatic gene expression and sphingolipid content also

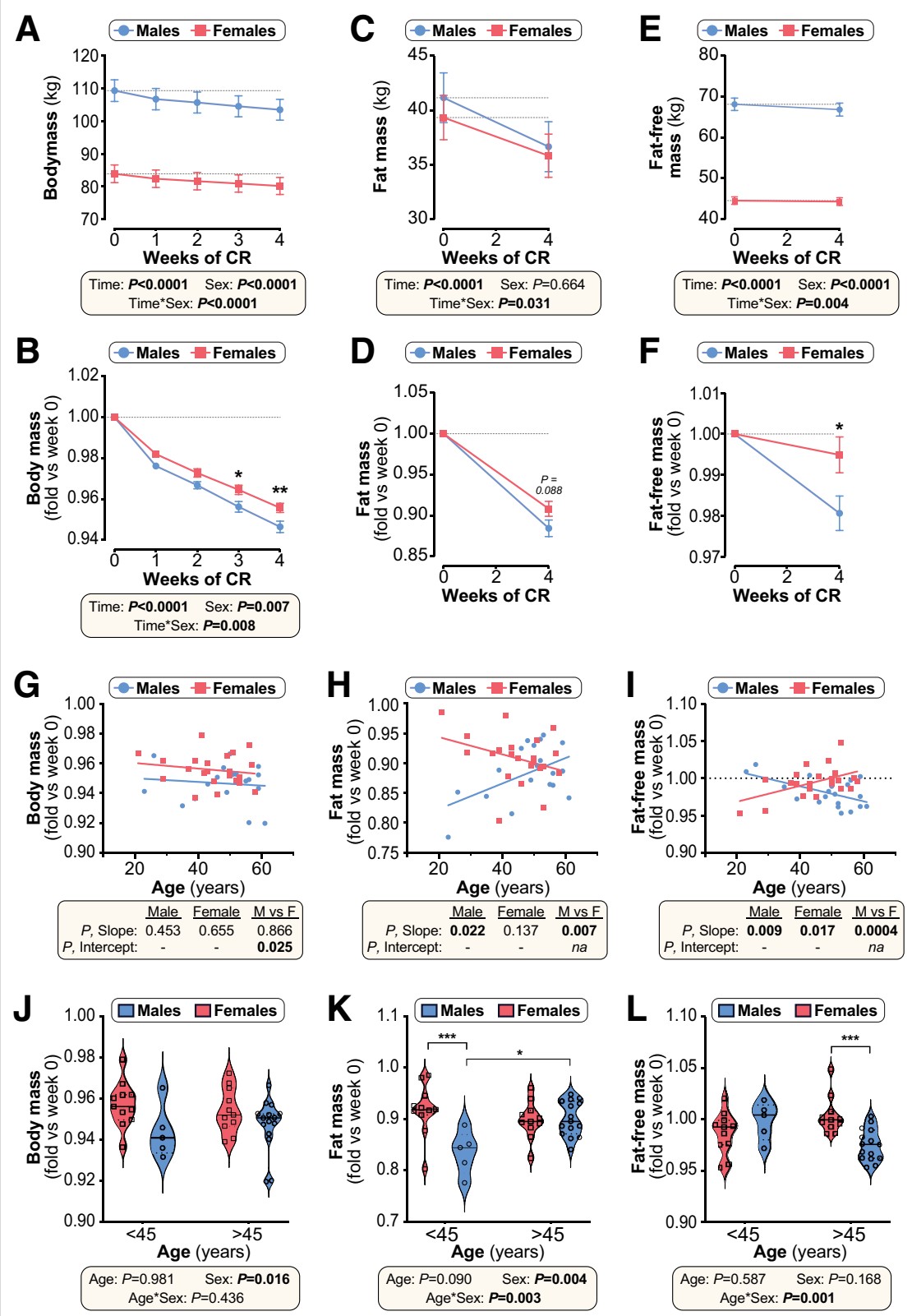

**Figure 10.** Effects of CR on body mass and body composition in humans are sex- and age-dependent. Twenty male and 22 female volunteers participated in a weight loss study involving a 4-week dietary intervention. (**A–F**) Body mass was recorded each week (**A,B**). Fat mass (FM) and fat-free mass (FFM) were measured by air-displacement whole-body plethysmography at weeks 0 and 4 (**C–F**). Body mass, fat mass, and fat-free mass are shown as absolute masses (**A,C,E**) or fold-change relative to baseline (**B,D,F**). Data are presented as mean ± SEM. For (**A–C**) and (**E**), significant effects of time,

*Figure 10 continued on next page*

*Figure 10 continued*

sex, and time*sex interaction were assessed using two-way ANOVA. In (**B**), (**D**) and (**F**), significant differences between males and females at each time point were determined by Šídák's multiple comparisons test (**B**) or unpaired T-test (**D,F**) and are indicated by * (p<0.05) or ** (p<0.01). (**G–I**) Simple linear regression of age vs fold-change (*week 4 vs week 0*) in body mass (**G**), fat mass (**H**) and fat-free mass (**I**). For each sex, significant associations between age and outcome (fold-change) are indicated beneath each graph as '*P, Slope*'. ANCOVA was further used to test if the age-outcome relationship differs significantly between males and females. ANCOVA results are reported beneath each graph as '*P, Slope*' and '*P, Intercept*' for males vs females (M vs F). In (**G**), similar slopes but different intercepts show that sex significantly influences weight loss, but the influence of age does not differ between the sexes. In (**H,I**) the slopes differ significantly, indicating that the age-outcome relationship differs between the sexes. (**J–L**) Fold-change (week 4 vs week 0) in body mass (**J**), fat mass (**K**) and fat-free mass (**L**) for males vs females separated into younger (<45 years) and older (>45 years) groups. Data are presented as violin plots overlaid with individual data points. Significant effects of age, sex, and age*sex interaction were assessed using two-way ANOVA with Tukey's multiple comparisons test. Overall p values for each variable, and their interactions, are shown beneath each graph. Significant differences between comparable groups are indicated by * (p<0.05) or *** (p<0.001). Source data are provided as a Source Data file. See also *Table 2* and *Figure 10—figure supplements 1–3*.

The online version of this article includes the following source data and figure supplement(s) for figure 10:

**Source data 1.** Effects of CR on body mass and body composition in humans are sex- and age-dependent.

**Figure supplement 1.** Baseline fat mass or BMI do not influence sex differences in the effects of CR on body mass or body composition.

**Figure supplement 1—source data 1.** Baseline fat mass or BMI do not influence sex differences in the effects of CR on body mass or body composition.

**Figure supplement 2.** CR-induced fat loss in mice is age- and sex-dependent.

**Figure supplement 2—source data 1.** CR-induced fat loss in mice is age- and sex-dependent.

**Figure supplement 3.** Body fat percentage in human CR participants.

**Figure supplement 3—source data 1.** Body fat percentage in human CR participants.

differ between young male and female mice, highlighting the liver as a key mediator of CR's sexually dimorphic metabolic effects. Finally, we reveal that humans also exhibit age-dependent sex differences in the CR response, with younger age groups showing greater fat loss in males than females whereas, with increasing age, CR-induced fat loss becomes similar between the sexes. Together, these data identify sex and age as factors that have a major influence on the metabolic health outcomes of CR.

## Age-dependent sex differences in CR's metabolic benefits

Our observation that young females resist overall weight loss during CR is consistent with some previous findings in mice (*Shi et al., 2007*; *Cawthorn et al., 2016*; *Piotrowska et al., 2016*), monkeys (*Mattison et al., 2017*), and humans (*Musante, 1976*; *O'Neil et al., 1979*; *Mauriège et al., 1999*; *Wong et al., 2012*; *Azar et al., 2016*; *Das et al., 2017*). Young females' resistance to CR-induced fat loss has also been reported in mice (*Shi et al., 2007*; *Cawthorn et al., 2016*; *Cawthorn et al., 2014*; *Dionne et al., 2016*; *Li et al., 2010*) and humans (*Redman et al., 2007*; *Mauriège et al., 1999*; *Leenen et al., 1993*; *Fontana et al., 2010*; *Evans et al., 2012*); however, fewer studies have directly assessed sex differences in the effect of CR on glucose homeostasis and insulin sensitivity. Many studies have included males and females but did not sex-stratify their analyses, thereby overlooking the potential impact of sex on these important metabolic outcomes (*Larson-Meyer et al., 2006*; *Kraus et al., 2019*). Nevertheless, CR in monkeys and humans has been found to improve glucose homeostasis and insulin sensitivity more so in males than in females (*Mattison et al., 2017*; *Wong et al., 2012*), consistent with our findings in young mice.

Our observations fundamentally extend this previous research by demonstrating, for the first time, that these sex differences are age dependent. The novelty of this finding highlights that relatively few studies have investigated CR when initiated in aged subjects. In 2004, Dhahbi et al showed that CR's health benefits persist even when initiated in aged mice (*Dhahbi et al., 2004*), and subsequent research has compared the metabolic effects of CR when implemented in young vs aged animals (*Sheng et al., 2021*; *Rohrbach et al., 2007*). Unfortunately, these studies did not include females, an oversight that has persisted in the CR field (*Figure 1*, *Figure 1—figure supplement 1*). Other notable research has comprehensively compared aged males and females, but only in the context of lifelong CR that was initiated in younger mice (*Mitchell et al., 2016*). Two recent studies have also highlighted age-dependent sex differences in the effects of methionine restriction or time-restricted

feeding, but these dietary interventions were limited to young vs middle-aged mice (*Forney et al., 2020*; *Chaix et al., 2021*). The most relevant previous research is perhaps from Brandhorst et al, who observed significant visceral fat loss in female mice fed a 'fasting-mimicking diet' from 16 to 20.5 months of age (*Brandhorst et al., 2015*). While this is consistent with our observations in aged females (*Figure 8*, *Figure 8—figure supplement 1*), it is limited by having not assessed glucose homeostasis nor including male counterparts for direct comparison. Thus, our study is the first to compare CR when initiated in young or old males and females and to thereby identify age-dependent sex differences in CR's metabolic effects.

Our findings have several implications for interpreting previous CR research. Although most studies show that young females resist overall weight loss and fat loss during CR, such sex differences have not been universally observed. For example, in young female rats some studies report that CR decreases body mass and/or fat mass to a similar extent in both sexes (*Porter et al., 2004*; *Valle et al., 2005*; *Valle et al., 2007*), while others find fat loss to be even greater in females than in males (*Hill et al., 1986*; *Hill et al., 1985*). CR-induced fat loss has also been noted in young female mice (*Fenton et al., 2009*; *Varady et al., 2010*; *Harvey et al., 2013*; *Zgheib et al., 2014*), albeit in studies that lack males for direct comparison. Several human studies also show no differences in total fat loss with CR, even in younger women who are unlikely to be menopausal (*Wong et al., 2012*; *Das et al., 2017*; *Janssen and Ross, 1999*). What are the reasons for these inconsistent results, and can they further inform the basis for sex differences in the CR response?

Considering our present findings, one possibility is that, even in younger subjects, age differences might have influenced the outcomes of these CR studies. Indeed, both Janssen et al and Das et al studied women who, while pre-menopausal, were significantly younger than their male counterparts (*Das et al., 2017*; *Janssen and Ross, 1999*). Given that males' fat loss decreases as they get older (*Figure 10H and K*), this disparity could have undermined the ability of these studies to detect sex differences in fat loss. A related, broader issue is that much human CR research has studied females who are over 45 years old (*Musante, 1976*; *Mauriège et al., 1999*; *Wong et al., 2012*; *Leenen et al., 1993*; *Evans et al., 2012*; *Wirth and Steinmetz, 1998*; *Parker et al., 2002*; *Weiss et al., 2006*). This is important because it is the age at which oestrogen levels typically begin to decline (*Diaz Brinton, 2012*) and after which we demonstrate no sex differences in fat loss (*Figure 10K*). Thus, our findings underscore the importance of properly age-matching males and females, and ideally confirming menopausal status, if the response to CR is to be reliably assessed.

While age is clearly an important variable, many animal studies still find no resistance to fat loss in young females (*Fenton et al., 2009*; *Varady et al., 2010*; *Harvey et al., 2013*; *Zgheib et al., 2014*), even when appropriately age-matched to male counterparts (*Porter et al., 2004*; *Valle et al., 2005*; *Valle et al., 2007*; *Hill et al., 1986*; *Hill et al., 1985*). Therefore, other factors must also be influencing CR's sexually dimorphic effects. One possibility is that males show a greater CR response because they are usually heavier and fatter than females at the onset of CR, and thus have more fat available to lose. Consistent with this 'baseline adiposity' hypothesis, during CR in rats Hill et al found total fat loss to be greater in young females than young males and, unlike in mice, pre-CR adiposity was greater in the females (*Hill et al., 1986*; *Hill et al., 1985*). Moreover, we show that sex differences in baseline adiposity become less pronounced with ageing, such that % fat mass is similar in aged male and female mice (*Figure 8D*). Therefore, this hypothesis could explain why CR decreases fat mass in the fatter, aged females, whereas young, leaner females resist this effect. However, three key points argue against this hypothesis. Firstly, we show that young females' resistance to fat loss still occurs in obese and overweight humans, among whom % fat mass is greater in females than in males, irrespective of age (*Figure 10—figure supplement 3*). This is consistent with numerous previous studies of overweight or obese humans, in which CR causes more weight loss and/or fat loss in males than females (*Redman et al., 2007*; *Musante, 1976*; *O'Neil et al., 1979*; *Mauriège et al., 1999*; *Wong et al., 2012*; *Azar et al., 2016*; *Leenen et al., 1993*; *Evans et al., 2012*; *Wirth and Steinmetz, 1998*). Secondly, in our human study we show that the sex differences in loss of fat or fat-free mass are unrelated to baseline adiposity or BMI (*Figure 10—figure supplement 1*). Finally, in young, non-obese humans, sex differences in fat loss persist even after controlling for variation in baseline fat and body mass (*Das et al., 2017*). Thus, sex differences in the CR response are a not simply a consequence of baseline differences in these parameters.

A third possibility is that young females' resistance to weight and fat loss can be overcome if the extent and/or duration of CR is great enough. Our CR protocol involves a 30% decrease in daily caloric intake over 6 weeks. In contrast, those studies reporting fat loss in young females typically used CR of a greater extent and/or duration. For example, Hill et al studied 60% CR for 11 weeks (*Hill et al., 1986*; *Hill et al., 1985*) while Valle et al used 40% CR for 14 weeks (*Valle et al., 2005*; *Valle et al., 2007*). Others have tested CR of a similar extent to our studies (20–30%) but for longer durations, ranging from 10 weeks to 6 months (*Porter et al., 2004*; *Fenton et al., 2009*; *Harvey et al., 2013*). It is now well established that the CR response is dose and duration dependent (*Speakman and Mitchell, 2011*; *Cawthorn et al., 2016*; *Dionne et al., 2016*; *Mitchell et al., 2016*; *Chen et al., 2015*; *Zhu et al., 2004*), with the former having been systematically investigated in male mice (*Mitchell et al., 2015b*; *Mitchell et al., 2015a*; *Mitchell et al., 2017*; *Derous et al., 2018*). Thus, a key goal for future studies will be to determine how sex and age influence these dose- and duration-dependent effects.

A final important consideration is the distinction between total versus regional adiposity. Indeed, the CALERIE trial finds no sex differences in the loss of total fat mass but does show that, compared to men, women resist loss of trunk fat during CR (*Das et al., 2017*). This is consistent with our data showing that fat loss in young male mice is greatest for the visceral depots (*Figure 2G*, *Figure 2—figure supplement 1C*, *Figure 2—figure supplement 2A*, *Figure 2—figure supplement 4G*), and with previous observations for CR in obese humans (*Redman et al., 2007*; *Leenen et al., 1993*; *Evans et al., 2012*; *Wirth and Steinmetz, 1998*). Some CR studies may therefore have failed to detect sex differences because they measured only total adiposity, missing out on possible site-specific differences (*Wong et al., 2012*; *Ballor and Poehlman, 1994*). Thus, any studies seeking to fully understand sex differences in CR's metabolic effects, whether in preclinical models or in humans, should always aim to assess adiposity on both whole-body and regional levels.

## Basis for young females' resistance to weight loss and fat loss

Our studies also shed light on the mechanistic basis for these sex differences. Previous murine CR studies show that females decrease total energy expenditure more than males, possibly resulting from greater suppression of BAT activity (*Shi et al., 2007*; *Valle et al., 2005*; *Valle et al., 2007*). Consistent with this, we find that CR suppresses energy expenditure more in young females than in young males, at least during the first week of CR (*Figure 3A–B*). In contrast, CR males and females have similar energy expenditure at week 3 of CR (*Figure 3A–B*), similar to our previous findings at week 4 of CR (*Suchacki et al., 2022*). Consistent with this, BAT activity is similar between AL and CR mice after 6 weeks of CR (*Figure 5*). Therefore, if using indirect calorimetry to understand the CR response, it is important to ensure that the calorimetry analyses coincide with timepoints at which the relevant CR responses occur.

Another advance over our previous calorimetry studies (*Suchacki et al., 2022*) is that, by using the higher-sensitivity Promethion system, we were able to measure RER and thereby identify sex differences in lipid metabolism. We find that, compared to CR males, CR females have greater postprandial lipogenesis but lower FA oxidation and lipolysis. This confirms previous observations in male or female mice only (*Bruss et al., 2010*; *Varady et al., 2010*) but extends these earlier findings by demonstrating key sex differences in these effects. It is also like the situation in humans, in which CR enhances WAT lipolysis in both sexes and females resist this effect (*Mauriège et al., 1999*). Together, these differences in lipid metabolism likely explain why young females resist fat loss during CR. Whether ageing alters these sex differences in lipid metabolism remains to be determined.

A final consideration is how time of CR feeding influences energy homeostasis. In our morning-fed mice, CR decreases energy expenditure more at nighttime than in the day. In contrast, Dionne et al found that in nighttime-fed mice CR decreases energy expenditure in the day but not at night (*Dionne et al., 2016*). However, both we and Dionne et al show that CR decreases 24 hr energy expenditure, demonstrating that this overall CR effect occurs irrespective of time of feeding.

## Importance of visceral fat loss for CR's metabolic effects

These effects on fat mass and lipid metabolism likely contribute to young females' resistance to improved glucose homeostasis during CR. Human studies have shown that inefficient WAT lipolysis is associated with impaired glucose metabolism in women (*Arner et al., 2018*) and that, during CR,

decreased fat mass is linked to the improvements in insulin sensitivity (*Larson-Meyer et al., 2006*). Similarly, studies of CR or lipectomy in animal models suggest that visceral fat loss contributes to CR's broader metabolic benefits (*Barzilai et al., 1998*; *Gabriely et al., 2002*; *Muzumdar et al., 2008*; *Tran et al., 2008*; *Larson-Meyer et al., 2006*). However, no previous research has investigated if this relationship between fat loss and insulin sensitisation differs between the sexes. Our work therefore supports and extends these previous studies by demonstrating that CR-induced fat loss coincides with improved glucose homeostasis and insulin sensitivity in males and aged females, whereas young females' resist these metabolic effects.

## The liver as a key driver of sex differences in glucose tolerance

Several of our findings highlight the importance of the liver as a key mediator of CR's sexually dimorphic metabolic effects. Firstly, our PET/CT data show that CR improves glucose tolerance without any marked increases in peripheral glucose uptake (*Figure 5*). This includes the BMAT-rich distal tibia, suggesting that, although BMAT has high basal glucose uptake (*Suchacki et al., 2020*), increased BMAT glucose uptake is not required for improved glucose tolerance during CR. Moreover, although CR does increase glucose uptake in WAT and the kidneys, this effect does not differ between males and females and therefore is unlikely to explain the sex differences in glucose tolerance (*Figure 5B*). Together our data are consistent with human studies showing that, in the fasted state, CR does not affect overall glucose disposal (*Gazdag et al., 1999*; *Johnson et al., 2016*); however, CR enhances insulin-stimulated glucose uptake (*Sequea et al., 2012*) and it is possible that this effect differs between the sexes. Our second relevant finding is that, during an OGTT, CR decreases the tAUC but not the iAUC, highlighting decreases in fasting glucose, rather than insulin-stimulated glucose disposal, as the main driver of the improvements in glucose tolerance. Thirdly, we show that CR's effects on hepatic gene expression differ between young male and female mice (*Figure 7*, *Figure 7—figure supplement 1*). These transcriptional differences suggest that, during CR, both sexes increase hepatic FA oxidation, but only males increase TCA cycle activity and oxidative phosphorylation (*Figure 7B–C*). Strikingly, females have higher plasma ketone concentrations, a marker of hepatic acetyl-CoA content (*Perry et al., 2017*). Based on these data, our model is that CR males utilise hepatic acetyl-CoA to support the TCA cycle and ATP generation via oxidative phosphorylation; in contrast, in CR females the hepatic acetyl-CoA accumulates, thereby stimulating pyruvate carboxylase and promoting gluconeogenesis (*Figure 7D*). Consequently, females are better at resisting hypoglycaemia during CR. As discussed below, future studies using metabolic tracers or related approaches will be important to further test these conclusions.

How is loss of visceral fat linked with these hepatic effects? Visceral adiposity is strongly associated with increased hepatic fat storage (*Liu et al., 2021*), and decreased liver fat coincides with improved insulin sensitivity during CR, both in rats and humans (*Perry et al., 2018*; *Taylor et al., 2018*). Moreover, young female mice resist the decreases in hepatic triglycerides following short-term fasting (*Della Torre et al., 2018*) and prolonged time-restricted feeding (*Piotrowska et al., 2016*). Therefore, we were surprised to find that CR does not affect hepatic triglyceride content in young mice of either sex (*Figure 6A*). This may be because our young AL controls are lean and healthy, and therefore already have livers that are not excessively fatty. In contrast, many of the previous studies investigated CR in animals or humans that are older and/or with type 2 diabetes (*Piotrowska et al., 2016*; *Perry et al., 2018*; *Taylor et al., 2018*), thereby increasing the baseline/control hepatic fat content. This would not explain why our findings differ to those of Della Torre et al, who studied fasting in young, healthy mice (*Della Torre et al., 2018*); however, it is perhaps unsurprising that our 6-week CR elicits different effects to those induced during shorter term fasting.

Our above observations are important because they show that CR can exert metabolic benefits without decreasing hepatic triglyceride content. In contrast, we find that effects on sphingolipids may play a role, with CR altering hepatic sphingolipid content more so in young males than in young females. Here, the effect of diet and/or sex depends on the sphingolipid species concerned (*Figure 6*; *Figure 6—figure supplement 1*). This is reminiscent of the findings of Norheim et al, who report sex differences in hepatic ceramide content in mice fed obesogenic diets (*Norheim et al., 2018*). They further showed that the relationship between ceramides, body fat and insulin sensitivity also differs between males and females, focussing on 16:0 ceramide (C16:0). Indeed, ceramides' influence on insulin sensitivity depends on the ceramide species. For example, hepatic increases in C16:0

have been implicated with the development of obesity-associated insulin resistance (*Chaurasia et al., 2019*; *Turpin et al., 2014*; *Raichur et al., 2014*; *Raichur et al., 2019*). This is seemingly at odds with our finding that CR increases hepatic C16:0 in young males but not young females (*Figure 6—figure supplement 1*). Thus, unlike in obesity, increased hepatic C16:0 during CR appears to be associated with improvements in insulin sensitivity.

The rate of conversion of ceramides to DHCs may be a root cause of insulin resistance (*Holland et al., 2007*; *Zhang et al., 2012*). This underscores the concept that the ratio between each ceramide and DHC species may influence metabolic function more than the absolute values of either (*Apostolopoulou et al., 2020*). More research is needed to determine how specific ceramide and DHC species, and the ratios thereof, impact sex differences in the health benefits of CR. Despite these remaining questions, our work extends previous observations by revealing that CR's hepatic effects are sexually dimorphic and that this contributes to the sex differences in glucose tolerance and insulin sensitivity.

## Oestrogen as a modulator of the CR response

In many cases, young females' resistance to the metabolic effects of CR is associated with them already being relatively metabolically healthy on an AL diet. For example, young AL females have lower visceral fat mass and are more glucose tolerant than AL males (*Figures 2G and 4B–C*), and are similar to CR males and females in terms of fasting insulin and HOMA-IR (*Figure 4D–E*). We also show that, for most sphingolipid species, the hepatic ceramide:DHC ratio is similar between AL females and CR mice of either sex (*Figure 6—figure supplement 1C*), while our RNA-seq data show that CR males are more similar to AL females than AL males (*Figure 7A*). These patterns suggest that the mechanisms through which CR decreases visceral adiposity, improves glucose tolerance and modulates sphingolipid metabolism are already at least partially active in young females, even on an AL diet.

This observation helps to clarify potential mechanisms for CR's age-dependent sex differences. Sexual dimorphism in metabolic function typically results from sex hormones, with genetic factors and/or developmental programming also playing a role. For example, X-chromosome-linked genetic factors influence sex differences in adiposity independently of sex hormones or other secondary sexual characteristics (*Mauvais-Jarvis et al., 2017*; *Link et al., 2020*), while the developmental and neonatal environment causes sexually dimorphic metabolic programming (*Dearden et al., 2018*). However, it is unlikely that these developmental or genetic factors explain the sex differences we observe. Developmental programming is unlikely because the early life environment of our mice was the same for both sexes and was devoid of any adverse or stressful stimuli. Similarly, if X-linked genetic factors were responsible then the sex differences would be expected to persist in aged females, but they mostly do not. This leaves sex steroids as the likely mediator. Consistent with this, oestrogen is now well known to influence metabolic function and the fasting response (*Mauvais-Jarvis, 2018*; *Della Torre et al., 2018*), and many actions of oestrogens overlap with those central to the effects of CR (*Shen et al., 2014*; *Wei and Huang, 2019*; *Gupte et al., 2015*; *Klinge, 2020*). Indeed, our RNA-seq data show that CR activates oestrogen response pathways in males and females (*Figure 7C*), further suggesting commonalities in the effects of oestrogens and CR. Herein, we further show that the sex differences in CR's metabolic effects are absent or greatly diminished when CR is initiated at 18 months of age, when C57BL/6 females are typically anoestrus (*Gee et al., 1983*; *Diaz Brinton, 2012*). Similarly, CR does decrease fat mass in young ovariectomised females (*Wheatley et al., 2011*), while oestrogen treatment protects male mice from obesity-induced insulin resistance (*Dakin et al., 2015*; *Zhu et al., 2014*). Therefore, we propose that young females resist the metabolic effects of CR because, through endogenous oestrogen action, they already have high activation of pathways that mediate CR's metabolic effects.

## Limitations and further considerations

Our study represents a significant advance in knowledge by revealing that age and sex have a substantial influence over the metabolic benefits of CR. This highlights the need for preclinical and human CR research to address these variables as key determinants of the CR response, a consideration that we show has been lacking in most CR research. We show that these sex differences occur on a systemic, physiological level and that they extend to cellular and molecular effects within adipose tissue and the liver, observations that identify potential mechanisms for the age-dependent sex differences;

however, a general limitation of our study is that these mechanisms remain to be directly addressed. Specific limitations are as follows.

Firstly, we used a feeding regimen common in rodent CR studies, with mice receiving a single ration of diet in the morning. We found that the sex differences persisted when the daily ration was given in the evening, demonstrating that the sexual dimorphism is not specific to a morning-fed regimen. However, such single-ration protocols invoke extensive daily fasting, with mice typically consuming all of the food within ~2 hr (*Acosta-Rodríguez et al., 2022*; *Acosta-Rodríguez et al., 2017*). Therefore, they include elements of both CR and of intermittent fasting. Recent studies have shown that such prolonged daily fasting contributes to the metabolic benefits and lifespan extension associated with CR, at least in male mice (*Acosta-Rodríguez et al., 2022*; *Pak et al., 2021*; *Mitchell et al., 2019*). Thus, it will be interesting to determine if these beneficial effects of prolonged fasting, and not just CR per se, also differ between males and females.

Secondly, regarding insulin sensitivity and glucose homeostasis, it would be informative to use hyperinsulinaemic-euglycaemic clamps and labelled metabolic tracers to comprehensively assess systemic and hepatic insulin sensitivity and determine if, as we propose, the differences in glucose homeostasis relate primarily to effects on gluconeogenesis and hepatic acetyl-CoA metabolism.

Thirdly, the role of oestrogen remains to be directly tested. This could be done in mice through ovariectomy studies and/or manipulation of ERα activity. We have also tried to determine oestradiol concentrations in our mouse plasma samples using mass spectrometry, a method required for reliable assessment of sex steroid concentrations (*Handelsman and Wartofsky, 2013*). However, despite extensive mass spectrometry optimisation efforts, we have not yet been able to reliably measure oestradiol using the relatively small plasma volumes available. Related to this, in our human studies we were unable to confirm the menopausal status of the female participants. Other limitations of our human studies are that they were done only in overweight or obese individuals; included relatively few younger participants; and lacked analysis of regional adiposity (e.g. visceral versus subcutaneous adipose tissue). These limitations reflect that our human study was not originally designed to directly test the influence of age and sex, with the data instead analysed retrospectively to address this. Future human research must therefore prioritise assessment of menopausal status and regional adiposity, both in lean and overweight/obese subjects, if the outcomes of CR are to be fully interpreted.

A fourth limitation regards the underlying molecular mechanisms. Pathways implicated in mediating CR's effects include activation of AMPK, SIRT, IGF-1 and FGF21, modulation of mitochondrial function and inhibition of mTOR (*Speakman and Mitchell, 2011*; *Kane et al., 2018*). Thus, genetic, pharmacological and/or other approaches to manipulate these candidate pathways could be used to further dissect the molecular mechanisms underlying CR's age-dependent sexually dimorphic metabolic effects. Such approaches are a promising basis for future research.

Beyond these specific limitations, a broader question is whether other effects of CR also differ between males and females in this age-dependent manner. Some research suggests that CR's effects on cardiovascular function and healthy ageing differ between males and females (*Kane et al., 2016*; *Ferland et al., 2016*; *Austad and Bartke, 2015*); however, given the dearth of CR studies that directly assess sex differences (*Figure 1*, *Figure 1—figure supplement 1*) there remains a critical need to determine how age and sex influence the many other outcomes of CR. This is important not only for understanding fundamental biology, but also if we are to translate the benefits of CR, or CR mimetics, to improve human health. For example, if younger women resist many of CR's health benefits, should any CR-based intervention focus only on older women, who may be more responsive?

Finally, while it will be useful to understand the extent and basis for these sex differences, it is also important to consider their broader purpose. Adipose tissue profoundly influences female reproductive function, with adipose-secreted factors acting within the central nervous system to indicate if the body's energy stores are sufficient to support reproduction (*Bohler et al., 2010*). The hormone leptin is the key adipose-derived factor that impacts the hypothalamic-pituitary-gonadal axis: if fat mass is too low, for example in anorexia nervosa or lipodystrophy, then leptin levels are also insufficient, resulting in decreased gonadotropin secretion and impaired reproductive capacity. Normal menses resumes in anorexic individuals who regain weight and fat mass, or in lipodystrophic individuals following leptin treatment (*Bohler et al., 2010*; *Misra and Klibanski, 2014*). Notably, we previously demonstrated that female mice resist hypoleptinaemia during CR (*Cawthorn et al., 2016*). Thus, we speculate that females resist fat loss during CR to preserve reproductive function. Given the tight interplay between

adipose tissue and systemic metabolic homeostasis, such resistance to fat loss then has consequences for CR's other metabolic effects. This evolutionary pressure to overcome periods of food scarcity would thereby have driven the divergence of the CR response in males and females, with implications for current efforts to exploit CR therapeutically to improve human health.

# Materials and methods

**Key resources table**

| Reagent type (species) or resource | Designation | Source or reference | Identifiers | Additional information |
|---|---|---|---|---|
| Chemical compound, drug | [18]F-Fluoro-deoxyglucose ([18]F-FDG) (radionuclide) | Edinburgh Clinical Research Imaging Centre (Edinburgh, UK) | N/A | |
| Chemical compound, drug | Ribozol | Amresco (USA) | N580 | |
| Commercial assay or kit | NEFA assay reagents | FUJIFILM Wako Chemicals Europe GmbH (Neuss, Germany) | 434–91795 436–91995 270–77000 | |
| Antibody | anti-HSL (Rabbit polyclonal) | Cell Signaling Technology (Danvers, MA, USA) | #4107 RRID:AB_2296900 | (1:1000) |
| Antibody | Anti-Phospho-HSL (Ser563) (Rabbit polyclonal) | Cell Signaling Technology (Danvers, MA, USA) | #4139 RRID:AB_2135495 | (1:1000) |
| Antibody | Anti-beta actin (Rabbit polyclonal) | Abcam (Cambridge, UK) | #ab8227 RRID:AB_2305186 | (1:1000) |
| Antibody | IRDye 800CW Goat anti-Rabbit IgG Secondary Antibody | LI-COR (Lincoln, NE, USA) | #926–32211 | (1:5000) |
| Strain, strain background (*Mus musculus*) | C57BL/6 J or C57BL/6NCrl | Charles River | 027 RRID:IMSR_CRL:027 | |
| Software, algorithm | Prism | GraphPad Software, LLC | v9.5.1 | |
| Software, algorithm | PMOD | PMOD Technologies LLC (Zurich, Switzerland) | v3.806 | |
| Software, algorithm | TrimGalore | https://www.bioinformatics.babraham.ac.uk/projects/trim_galore/ | V0.6.6 | |
| Software, algorithm | FastQC | https://www.bioinformatics.babraham.ac.uk/projects/fastqc/ | v0.11.7 | |
| Software, algorithm | STAR | *Dobin et al., 2013* | v2.7.10a | |
| Software, algorithm | Subread | *Liao et al., 2014* | v1.5.2 | |
| Software, algorithm | R | https://www.r-project.org/ | v4.2.0 | |
| Software, algorithm | RStudio | https://posit.co/download/rstudio-desktop/ | v2022.12.0+353 | |
| Software, algorithm | DESeq2 | *Love et al., 2014* | v1.36.0 | |
| Software, algorithm | GSEA | *Mootha et al., 2003*; *Subramanian et al., 2005* | V4.3.2 | |
| Software, algorithm | Morpheus | https://software.broadinstitute.org/morpheus | NA | |
| Software, algorithm | Enricher | *Xie et al., 2021*; *Kuleshov et al., 2016*; *Chen et al., 2013* | NA | |
| Software, algorithm | appyters | *Clarke et al., 2021* | NA | |
| Software, algorithm | Image Studio | LI-COR Biosciences (USA) | v5.2 | |
| Software, algorithm | ExpeData | Sable Systems International (Las Vegas, USA) | v1.9.27 | |

## Animals

Studies in C57BL/6NCrl and C57BL/6JCrl mice were approved by the University of Edinburgh Animal Welfare and Ethical Review Board and were done under project licenses granted by the UK Home Office. Mice were originally obtained from Charles River (UK) and then bred and aged in-house. They were housed on a 12 hr light/dark cycle in a specific-pathogen-free facility with free access to water and food, as indicated. Data herein are reported according to the ARRIVE guidelines (*Percie du Sert et al., 2020*); no checklist is provided, but key details are as follows. Differences in mouse cohorts are summarised in *Table 1*, along with the groups being compared and the experimental unit. The exact number of mice used is stated in the figure legends. Sample sizes for each cohort were determined by power calculations using G*Power software, with effect sizes based on pilot studies and/or previously published data for fat loss and improved glucose tolerance (the primary outcomes being tested) (*Cawthorn et al., 2016*; *Cawthorn et al., 2014*); to increase power, data for young mice are reported across multiple cohorts from at least four separate studies, each of which revealed the same sex differences in CR's metabolic effects. Mice were randomly assigned to each diet. To minimise potential confounders, such as the order of treatments or measurements, mice were treated or analysed based on the order of their ID number, a variable randomly distributed across the experimental variables (diet and sex). Blinding was not possible during the in vivo experiments because the experimenter(s) can see, based on presence or absence of diet, which mice are AL and which are CR; however, data analysis was done in a blinded manner. Mice were excluded from the final analysis only if there were confounding technical issues or pathologies discovered at necropsy. Ten mice from the young study were excluded from the analysis of OGTT and necropsy tissue masses because of a technical issue during gavaging for the OGTT, which compromised their subsequent food intake. One mouse was excluded from PET/CT data analysis because it accidentally had two doses of FDG administered. One aged mouse was excluded because of developing confounding liver tumours. For some cohorts, specific tissues were not weighed at necropsy and therefore these mice have no data for these read-outs. For further details, please see the Source Data files.

## Mouse CR studies

Before assigning to AL or CR diets, mice were housed individually between 8 and 9 weeks of age (for young mice) or 77–78 weeks of age (for old mice). During this time, mice were fed a control diet (Research Diets, D12450B) ad libitum and daily food intake was determined by weighing the food remaining in each cage each day. Following 1 week of single housing, all mice were randomly assigned to control diet (Research Diets D12450B, provided AL) or the CR diet (Research Diets D10012703, administered at 70% of the average daily AL diet consumption). This CR diet is formulated for iso-nutrition at 70% CR, ensuring that malnutrition is avoided. The evening-fed CR mice (*Figure 3—figure*

**Table 1.** Summary of CR protocol in each group of mice.
Because mice are singly housed, each mouse represents an independent experimental unit.

|  | Young | Aged | Evening-fed |
|---|---|---|---|
| Age at single housing | 8 weeks | 77 weeks | 8 weeks |
| Age at start of CR | 9 weeks | 78 weeks | 9 weeks |
| Time of feeding | 0900–1000 | 0900–1000 | 1800–1900 |
| Duration of CR | 6 weeks | 6 weeks | 4 weeks |
| Related data | *Figures 2–7*, *Figure 2—figure supplements 1–3*; *Figure 3—figure supplements 1–2*; *Figure 5—figure supplement 1*; *Figure 6—figure supplement 1*; *Figure 7—figure supplement 1*; *Figure 10—figure supplement 2* | *Figures 8–9*; *Figure 8—figure supplement 1*; *Figure 10—figure supplement 2* | *Figure 2—figure supplement 4*; *Figure 4—figure supplement 1* |
| Experimental unit | Single mouse |  |  |
| Groups compared | AL vs CR (within sex); Male vs Female (within diet); and diet*sex interactions |  |  |

**Table 2.** Human subject characteristics at baseline.
Age, BMI, glucose and daily caloric intake (Diet) are mean ± SD. ND = not determined.

| Sex | Number of subjects | Age (years) | BMI (kg/m²) | Glucose (mM) | Diet (kcal/day) | Diabetic |
|---|---|---|---|---|---|---|
| Male | 20 | 48.61±10.7 | 34.7±4.2 | 5.6±0.7 | 1616±214 | 0% |
| Female | 22 | 44.7±9.7 | 32.8±3.7 | 5.2±0.3 | 1325±62 | 0% |

*supplement 1*) were fed daily between 1800 and 1900, while all other CR mice were fed daily between 0900 and 1000. We find that CR mice consume their entire ration of diet within 2–3 hr of feeding, consistent with other CR studies (*Acosta-Rodríguez et al., 2022*; *Acosta-Rodríguez et al., 2017*). For the first 2 weeks of CR mice were monitored and weighed daily to ensure they did not lose more than 30% of their initial body mass. For the following 4 weeks of CR, body mass was recorded weekly. Mice were assessed non-invasively for body fat, lean mass, and free fluid weekly from weeks 0 to 6 using TD-NMR (Minispec LF90II; Bruker Optics, Billerica, MA, USA). Blood ketone concentrations were measured from weeks 0 to 4 using a GKI-Bluetooth Blood Glucose & Ketone Meter (Keto-Mojo Europe, Amsterdam, Netherlands). At necropsy, mice were humanely sacrificed via cervical dislocation and decapitation. Tissues were harvested, weighed, and stored either in 4% formalin or on dry ice. Tissues in formalin were fixed at 4 °C for 2–3 days before being washed and stored in DPBS (Life Technologies). Tissues on dry ice were stored at –80 °C prior to downstream analysis.

## Human weight loss study – participants

This study was done before pre-registration was a requirement for clinical trials. Herein, we retrospectively analysed the study outcomes to investigate if sex and/or age influence the CR response. Forty-five overweight and obese males and females; age 21-61y, Body Mass Index (BMI) 26–42 kg/m², were recruited by newspaper advertisement to participate in a weight loss (WL) dietary intervention study. Thus, the participants were non-randomly selected individuals who were sufficiently motivated to actively respond to the request for volunteers. During recruitment, participants underwent a medical examination and their general practitioners were contacted to confirm medical suitability to participate in the study. Written, informed consent was obtained, and the study was reviewed by the NHS North of Scotland Research Ethics Service (ethics number 09 /S0801/80). Participants were informed that the study was funded by an external commercial sponsor but the company was not named, and all packaging of the food range was removed before distribution. Three participants withdrew for personal reasons therefore data for n=42 is presented throughout; participant characteristics are summarised in *Table 2*. No data points were excluded from the final analyses.

## Human weight loss study – protocol

Participants were asked to attend the Human Nutrition Unit (HNU) at the Rowett Institute of Nutrition and Health (Aberdeen, UK) twice weekly to pick up food supplies. Participants had their basal energy requirements determined and each participant was then fed an individualised diet with a caloric content equivalent to 100% of their resting metabolic rate (*Table 2*). This approach was taken to standardise the diet to account for individual energy requirements and energy restriction. All required food, drinks and snacks were provided and the study dietician was present to answer any questions. Participants were provided with a diet guide whereby women were encouraged to consume no more than 6.3 MJ/d (1500 kcal) and men 8.4 MJ/d (2000 kcal). They were encouraged to consume 3 meals a day (breakfast, lunch and dinner) with a portion of protein at each meal, and 5 portions of fruit and vegetables, 2 portions of fats, 3 portions of milk and dairy and 4 portions of wholegrain bread, potatoes, and cereals each day. Participants completed a descriptive food diary every day to record what and when they ate. Participants ate ad libitum, choosing when and what to eat from their supplies. They completed a detailed 7d weighed dietary record during week 4 of the WL phase of the study, to compare eating habits with a baseline 7d weighed record they completed before the study started. Energy and nutrient intakes were calculated using the Windiets programme (Univation Ltd; The Robert Gordon University, Aberdeen, UK).

## Human weight loss study – Measurement of body composition

Before and after WL, at weeks 0 and 4 respectively, measurements of body composition were conducted under standardized conditions as described previously (*Johnstone et al., 2005*), with participants instructed to fast overnight (at least 10 h). Height was measured at the beginning of the study to the nearest 0.5 cm using a stadiometer (Holtain Ltd, Crymych, Dyfed, Wales). The body weight of the participants was measured at screening, during food provision days and on all test days, wearing a previously weighed dressing gown, to the nearest 100 g on a digital scale (DIGI DS-410, C.M.S. Weighing Equipment Ltd, London, UK). Fat mass (FM) and fat free mass (FFM) were measured with the use of air displacement whole-body plethysmography (BodPod Gold – Body Composition System, Cosmed, Italy) (*Mccrory et al., 1995*).

## Indirect calorimetry

Male and female AL and CR mice were housed individually in Promethion CORE System cages (Sable Systems International, Las Vegas, USA) from 9 to 10 or 12–13 weeks of age, corresponding to weeks 0–1 ('week 1') or 2–3 ("week 3") after beginning AL or CR feeding. At each timepoint, mice entered the cages around 11 am on day 1 and were housed for three nights (four days in total). Data herein are from the 48 h period between 7am on day 2 and 7am on day 4, to allow habituation to the new cages during day 1. Before and after Promethion housing, each mouse was weighed and body composition determined by TD-NMR. Measurements of energy expenditure, oxygen consumption, carbon dioxide production, and physical activity were analysed using ExpeData software according to the manufacturer's instructions. FA oxidation was determined as described previously (*Bruss et al., 2010*).

## Oral glucose tolerance tests (OGTT)

Mice were given a half portion of food the night (6pm) prior to the metabolic challenge. At 12:00 pm on the day of the OGTT a basal blood sample was collected by venesection of the lateral tail vein into EDTA-coated capillary tubes (Microvette 16.444, Sarstedt, Leicester, UK) and the basal glucose levels was measured using a glucometer (OneTouch Verio IQ, LifeScan Inc, Zug, Switzerland). D-glucose (G8270, Sigma, Poole, UK) was then administered at 2 mg per g of body mass by oral gavage of a 25% (w/v) solution. At 15, 30, 60, and 120 minut post-gavage, blood glucose was measured by a glucometer and tail vein blood was sampled in EDTA tubes, as for the basal (0 min) timepoint. Blood samples were kept on ice prior to isolation of plasma. Insulin was then measured in plasma samples by ELISA, following the manufacturer's instructions (cat. #90080, ChrystalChem, Chicago, IL, USA). Insulin was converted from ng/ml to µU/mL by multiplying the ng/mL value by 28.7174 (based on mouse insulin molar mass of 5803.69). Homeostatic Model Assessment for Insulin Resistance (HOMA-IR) and Matsuda index were calculated as previously described (*Matthews et al., 1985*; *Matsuda and DeFronzo, 1999*).

## PET/CT scanning

Immediately before and 15- and 60 min post $^{18}$F-FDG intraperitoneal injection, blood glucose was measured by tail venesection and blood was sampled directly into EDTA-microtubes (Sarstedt, Leicester, UK). At 60 min post $^{18}$F-FDG injection, mice were anaesthetised and $^{18}$F-FDG distribution was assessed by PET/CT scanning using a nanoScan PET/CT 122 S scanner (Mediso, Budapest, Hungary). Following the PET/CT scan, mice were sacrificed by overdose of anaesthetic (isoflurane), death was confirmed by cervical dislocation and tissues of interest were then dissected in a lead-shielded area. Half of each of the dissected tissues were snap frozen on dry ice or placed into 10% formalin. $^{18}$F-FDG uptake was then determined using a Wizzard2 gamma counter (PerkinElmer, USA) as previously described (*Suchacki et al., 2020*).

## PET/CT analysis

PET/CT images were reconstructed and data analysed using PMOD version 3.806 (PMOD, Zurich, Switzerland). SUVs were calculated for regions of interest, namely BAT, iWAT, gWAT; heart; bone tissue (without BM) from tibiae, femurs, and humeri; To distinguish bone tissue from BM, a calibration curve was generated using HU obtained from the acquisition of a CT tissue equivalent material (TEM) phantom (CIRS, model 091) and mouse CT scans, as previously described (*Suchacki et al., 2020*).

## Histology and adipocyte size quantification

Dissected mouse tissue was fixed for 48 hr and paraffin embedded by the histology core at The University of Edinburgh's Shared University Research Facilities (SuRF). Paraffin-embedded tissue sections were then sectioned at 100 μm intervals using a Leica RM2125 RTS microtome and collected onto 76x26 mm StarFrost 624 slides (VWR, UK). The slides were baked at 37 °C overnight before Haematoxylin and Eosin (H&E) staining. Quantification of adipocyte number and size was performed on H&E-stained adipose tissue using the ImageJ plug-in Adiposoft, an automated software for the analysis of adipose tissue cellularity in histological sections (*Galarraga et al., 2012*).

## NEFA analysis in plasma samples

NEFA standard solution (Wako, 270–77000) was serially diluted to concentrations of 2, 1.5, 1, 0.5, 0.25, and 0.125 mM. The standard dilutions, blank sample and plasma (2 μL) were pipetted into separate duplicate wells of a 96-well plate. Reagent 1 (200 μL) (Wako, 434–91795) was added to each well and the plate incubated at 37 °C for 5 min. Absorbance of each well was measured at 550 and 660 nm. Reagent 2 (100 μL; Wako, 436–91995) was then added to each well and the plate was incubated again at 37 °C for 5 min and absorbance measured at 550 and 660 nm. The 660 nm reading was subtracted from the 550 reading from both measurements and the mean blank absorbance subtracted from the mean duplicate absorbance from each sample. NEFA concentrations were then determined by regression analysis.

## Protein isolation and western blotting

Frozen WAT samples were lysed in preheated SDS lysis buffer (1% sodium dodecyl sulfate (SDS), 0.06 M Tris pH 6.8, 12.7 mM ETDA, 1 mM PMSF, 1 mM sodium orthovanadate). After tissue disruption, the lysate was boiled at 95 °C on a dry heat block for 10 min. Boiled lysate was centrifuged at 16,000×$g$ and the cleared liquid fraction below the lipid layer and above the pellet was removed to a new tube for downstream analyses. Protein concentration was quantified using the BCA protein assay (Pierce 23225). For SDS polyacrylamide gel electrophoresis, lysates were diluted to equal protein concentration in lysis buffer plus 4 X SDS loading buffer (4% SDS, 240 mM Tris-HCL ph6.8, 40% glycerol, 0.276%(w/v) beta-mercaptoethanol, 0.05% Bromophenol blue). Dithiothreitol was added to the lysates (25 mM final concentration). Samples were boiled for 5 min, cooled on ice for 1 min, vortexed, and equal amounts of protein (13 μg per lane) separated on gradient polyacrylamide gels (Bio-Rad, Hercules, CA, USA). Samples were then transferred to Immobilon-FL transfer membrane (Millipore, Billerica, MA, USA). After transfer, membranes were blocked in 5% milk for 1 hr at RT. Membranes were incubated in primary antibodies (Key resources table; each in 5% BSA) overnight at 4 °C, followed by incubation with secondary antibody (Key resources table) at 1:5000 dilution for 1 hr at RT. Fluorescent color was detected with Odyssey CLx Imager (LI-COR, Lincoln, NE, USA). Band signal was quantified using Image Studio v5.2 (LI-COR, Lincoln, NE, USA). Quantified data obtained from two different membranes were combined by using the strength of the signal of the ladder bands as a control for intensity differences between the two membranes.

## Liver ceramide quantification

Liver ceramide and dihydroceramide content was determined using LC-MS (Lipidomics Core Facility; University of the Highlands and the Islands, Inverness, UK). As per (*Folch et al., 1957*), 100 mg of frozen liver tissue was homogenised in ice-cold MeOH using a mechanical tissue homogeniser. Fifty microlitres of each sample was combined with 100 μL ceramide 17:0 (1 μM in MeOH) and 25 μL dihydroceramide 12:0 (2 μM in MeOH; Avanti Polar Lipids, Alabaster, AL, USA) as internal standards. MeOH (1.875 mL) and CHCl$_3$ (4 mL) were added. Samples were then stored on ice for 1 hr with intermittent vortex mixing, centrifuged at 2000 rpm for 5 min at 4 °C and the supernatant was collected. KCl (0.9%; 1.45 mL per sample) was added to the supernatant and the samples were centrifuged at 2000 rpm for 5 min at 4 °C. The lower phase was collected and evaporated under nitrogen and reconstituted in 1 mL CHCl$_3$. Solid phase extraction was performed via a 100 mg/3 mL Isolute silica column (Biotage, Uppsala, Sweden). The column was conditioned twice with 3 mL CHCl$_3$ and then loaded with 1 mL of the sample. The bound sample was washed with 2 washes of 3 mL CHCl$_3$ then eluted with 2 additions of ethylacetate CHCl$_3$ (1:1 v/v). The collected eluate was evaporated and dried under vacuum and reconstituted in 200 μL MeOH. LC-MS/MS analyses were performed in positive ion mode

on a Thermo TSQ Quantum Ultra triple quadrupole mass spectrometer equipped with a HESI probe and coupled to a Thermo Accela 1250 UHPLC system. The ceramides and dihydroceramides were separated on a Kinetex C8 HPLC column (1.7 µm; 100×2.1 mm; catalogue number 00D-4499-AN, Phenomenex, Macclesfield, UK). Mobile phase A was 90% $H_2O$, 9.9% acetonitrile and 0.1% formic acid. Mobile phase B was 99.9% acetonitrile and 0.1% formic acid. The gradient was held at 80% mobile phase B for 1 min then gradually increased to 100% mobile phase B over 15 min. Mobile phase B at 100% was maintained for 1 min and then equilibrated to storage conditions. The flow rate used was 500 µL per min at 40 °C. Two µL of each sample was injected into the LC-MS/MS for ceramide analysis and 10 µL for dihydroceramide. Total ceramide and dihydroceramide concentrations were calculated from the summed concentrations of all the monitored molecular species. All values were normalised to wet weight of liver. Principal component analysis for ceramide and dihydroceramide data was done using RStudio (RStudio Team, USA), using the code provided (see 'Data and code availability', below).

## RNA-seq

RNA was isolated from tissues using Ribozol solution (cat. No. N580, Amresco, USA,) according to the manufacturer's protocol. RNA quantity and quality were quantified using a NanoDrop spectrophotometer (Thermo Fisher Scientific,USA) and RNA Integrity Number (RIN) was assessed using Agilent RNA 6000 Pico Kit and an Agilent 2100 bioanalyzer (Agilent, USA). Library preparation and sequencing was performed by BGI (Shenzhen, Guangdong, China); 100 bp pair-end sequencing was done using a DNBseq-G400 (MGI Tech). Calculations were done using Eddie Mark 3, the University of Edinburgh's high-performance computing cluster. After the trimming by TrimGalore and quality control by FastQC, sequences were aligned to the mouse genome (mm10) with the annotation data from the website of University of California Santa Cruz (https://hgdownload.soe.ucsc.edu) using STAR (v2.7.10a) (**Dobin et al., 2013**). Mapped reads were counted using featureCounts in Subread package (v1.5.2) (**Liao et al., 2014**) and subsequent analyses performed using R, RStudio, GSEA, Enricher, and Morpheus. Count data were normalised and analysed using DESeq2 (v1.36.0) to detect differentially expressed genes (**Love et al., 2014**). Principal component analysis was performed using the prcomp function for 500 genes with the highest variation among samples after transforming the raw count data using the vst function from the DESeq2 package. R code used for the analysis are provided (see 'Data and code availability", below). For GSEA, a gene set collection including eight gene sets was used for **Figure 7C**; further details are in **Table 3**. Heatmaps were drawn with Morpheus.

## Statistical analysis

Data were analysed for normal distribution using the Shapiro-Wilk normality test. Normally distributed data were analysed by ANOVA, mixed models or t-tests, as appropriate; mixed models were used for repeated-measures analyses in which some data points were missing or had to be excluded for some

**Table 3.** Gene sets used for GSEA.

Further details can be found by searching for the Standard Gene Set name at http://www.gsea-msigdb.org/gsea/msigdb/mouse/genesets.jsp.

| Gene Set name (Standard) | Systematic Name | Gene set name (shown in Figure 7C) |
|---|---|---|
| REACTOME_GLUCONEOGENESIS | MM15392 | GLUCONEOGENESIS |
| HALLMARK_OXIDATIVE_PHOSPHORYLATION | MM3893 | OXIDATIVE_PHOSPHORYLATION |
| REACTOME_CITRIC_ACID_CYCLE_TCA_CYCLE | MM15407 | CITRIC_ACID_CYCLE_TCA_CYCLE |
| HALLMARK_GLYCOLYSIS | MM3894 | GLYCOLYSIS |
| REACTOME_MITOCHONDRIAL_FATTY_ACID_BETA_OXIDATION | MM15462 | FATTY_ACID_BETA_OXIDATION |
| WP_FATTY_ACID_BIOSYNTHESIS | MM15885 | FATTY_ACID_BIOSYNTHESIS |
| HALLMARK_ESTROGEN_RESPONSE_EARLY | MM3872 | ESTROGEN_RESPONSE_EARLY |
| HALLMARK_ESTROGEN_RESPONSE_LATE | MM3873 | ESTROGEN_RESPONSE_LATE |

mice. Where data were not normally distributed, non-parametric tests were used. When appropriate, p values were adjusted for multiple comparisons. Data are presented as mean ± SEM or as Violin plots, as indicated in the figure legends. All statistical analyses were performed using Prism software (GraphPad, USA). A p-value<0.05 (after adjustment for multiple comparisons) was considered statistically significant.

## Acknowledgements

This work was supported by grants from the Medical Research Council (MR/M021394/1 to WPC), the British Heart Foundation (BHF) (RG/16/10/32375 and FS/19/34/34354 to AAST; 4 year BHF PhD Studentship to BJT and RJS), the Takeda Science Foundation (Fellowship for Young Japanese MDs & PhDs Studying Abroad, to YMI), the Japan Society for the Promotion of Science (JSPS Overseas Research Fellowship, to YMI), the Japan Foundation for Applied Enzymology (to YMI), the University of Edinburgh (Chancellor's Fellowship to WPC; PhD Studentship to AL), the Carnegie Trust (RIG007416 to WPC), the Wellcome Trust (Technology Development Award 221295/Z/20/Z to AAST; Multi-user equipment grant 223818/Z/21/Z for the Promethion system), the Chief Scientist Office of the Scottish Government (SCAF/17/02 to RHS), KAKENHI grants from MEXT/JSPS (21K08431 to HK), and a grant from the National Center for Global Health and Medicine (20A1010 to HK). CF and AMJ gratefully acknowledge financial support from the Scottish Government as part of the RESAS Strategic Research Programme at the Rowett Institute, University of Aberdeen. We thank Ami Onishi for assistance with indirect calorimetry studies using the Promethion system, and the BHF for supporting Ami Onishi's salary through the Centre for Research Excellence Award III (RE/18/5/34216). The BHF also provided funding towards establishment of the Edinburgh Preclinical PET/CT laboratory (RE/13/3/30183) and support of the 2018 Very Important Project (VIP) PET prize, for which we are grateful. Finally, we thank Dr Matthew Bennett (University of Edinburgh) for advice analysing RNA-seq data; Prof Robert Semple (University of Edinburgh) and Prof Ormond MacDougald (University of Michigan) for helpful advice to interpret our findings; Stefanie Fung (University of Edinburgh) for assistance isolating RNA for RNA-seq; Dr Hai-Bin Ruan (University of Minnesota Medical School) and Dr Eleonora Pagano (Pontifical Catholic University of Argentina) for kindly providing a full text of *Ballor and Poehlman, 1994* study (*Ballor and Poehlman, 1994*), and all staff at Edinburgh Bioresearch & Veterinary Services for their superb technical support. Finally, this manuscript was written entirely by the authors and without any use of large language models.

## Additional information

### Funding

| Funder | Grant reference number | Author |
| --- | --- | --- |
| Medical Research Council | MR/M021394/1 | William P Cawthorn |
| British Heart Foundation | RG/16/10/32375 | Adriana AS Tavares |
| British Heart Foundation | FS/19/34/34354 | Adriana AS Tavares |
| British Heart Foundation | 4-year PhD studentship | Benjamin J Thomas Richard J Sulston |
| Takeda Science Foundation | Fellowship for Young Japanese MDs & PhDs Studying Abroad | Yoshiko M Ikushima |
| Japan Society for the Promotion of Science | JSPS Overseas Research Fellowship | Yoshiko M Ikushima |
| Japan Foundation for Applied Enzymology | | Yoshiko M Ikushima |
| University of Edinburgh | Chancellor's Fellowship | William P Cawthorn |
| University of Edinburgh | PhD Studentship | Andrea Lovdel |

| Funder | Grant reference number | Author |
|---|---|---|
| Carnegie Trust for the Universities of Scotland | RIG007416 | William P Cawthorn |
| Wellcome Trust | 221295/Z/20/Z | Adriana AS Tavares |
| Chief Scientist Office of the Scottish Government Health Directorates | SCAF/17/02 | Roland H Stimson |
| KAKENHI | 21K08431 | Hiroshi Kobayashi |
| National Center for Global Health and Medicine | 20A1010 | Hiroshi Kobayashi |
| Scottish Government | RESAS Strategic Research Programme | Claire Fyfe
Alexandra M Johnstone |

The funders had no role in study design, data collection and interpretation, or the decision to submit the work for publication. For the purpose of Open Access, the authors have applied a CC BY public copyright license to any Author Accepted Manuscript version arising from this submission.

## Author contributions

Karla J Suchacki, Conceptualization, Data curation, Formal analysis, Funding acquisition, Investigation, Visualization, Methodology, Writing – original draft, Writing – review and editing; Benjamin J Thomas, Conceptualization, Data curation, Formal analysis, Investigation, Visualization, Methodology, Writing – original draft, Writing – review and editing; Yoshiko M Ikushima, Data curation, Formal analysis, Investigation, Visualization, Methodology, Writing – original draft, Writing – review and editing; Kuan-Chan Chen, Data curation, Formal analysis, Investigation, Visualization, Methodology; Claire Fyfe, Data curation, Formal analysis, Investigation, Methodology; Adriana AS Tavares, Resources, Funding acquisition, Methodology; Richard J Sulston, Conceptualization, Data curation, Formal analysis, Investigation; Andrea Lovdel, Data curation, Formal analysis, Investigation; Holly J Woodward, Xuan Han, Domenico Mattiucci, Eleanor J Brain, Carlos J Alcaide-Corral, Investigation; Hiroshi Kobayashi, Formal analysis, Methodology; Gillian A Gray, Resources, Supervision, Funding acquisition, Project administration, Writing – review and editing; Phillip D Whitfield, Formal analysis, Investigation, Methodology; Roland H Stimson, Resources, Supervision, Funding acquisition, Writing – review and editing; Nicholas M Morton, Resources, Formal analysis, Supervision, Funding acquisition, Methodology, Project administration, Writing – review and editing; Alexandra M Johnstone, Conceptualization, Resources, Data curation, Supervision, Funding acquisition, Investigation, Methodology, Project administration, Writing – review and editing; William P Cawthorn, Conceptualization, Resources, Data curation, Formal analysis, Supervision, Funding acquisition, Investigation, Visualization, Methodology, Writing – original draft, Project administration, Writing – review and editing

## Author ORCIDs

Karla J Suchacki ⓘ http://orcid.org/0000-0002-4688-4126
Benjamin J Thomas ⓘ http://orcid.org/0000-0001-8344-5256
Yoshiko M Ikushima ⓘ http://orcid.org/0000-0002-9987-874X
Kuan-Chan Chen ⓘ http://orcid.org/0000-0001-7653-6491
Claire Fyfe ⓘ http://orcid.org/0000-0001-6521-2651
Adriana AS Tavares ⓘ http://orcid.org/0000-0001-7505-9144
Andrea Lovdel ⓘ http://orcid.org/0000-0002-4262-8491
Carlos J Alcaide-Corral ⓘ http://orcid.org/0000-0002-3002-0649
Hiroshi Kobayashi ⓘ http://orcid.org/0000-0002-0924-1252
Gillian A Gray ⓘ http://orcid.org/0000-0003-3104-3305
Phillip D Whitfield ⓘ http://orcid.org/0000-0002-0814-4119
Roland H Stimson ⓘ http://orcid.org/0000-0002-9002-6188
Nicholas M Morton ⓘ http://orcid.org/0000-0001-8218-8462
Alexandra M Johnstone ⓘ http://orcid.org/0000-0002-5484-292X
William P Cawthorn ⓘ http://orcid.org/0000-0001-7832-5057

### Ethics

Written, informed consent was obtained, and the study was reviewed by the NHS North of Scotland Research Ethics Service (ethics number 09/S0801/80).

Studies in C57BL/6NCrl and C57BL/6JCrl mice were approved by the University of Edinburgh Animal Welfare and Ethical Review Board and were done under project licenses granted by the UK Home Office (project license numbers 708617 and PP2299608).

### Decision letter and Author response

Decision letter https://doi.org/10.7554/eLife.88080.sa1
Author response https://doi.org/10.7554/eLife.88080.sa2

## Additional files

### Supplementary files

• MDAR checklist

### Data availability

All source data and code from which the figures are based is available through University of Edinburgh DataShare at https://doi.org/10.7488/ds/3758 under a Creative Commons Attribution (CC-BY) licence. Raw data from RNA sequencing have been depositied at NCBI's GEO repository with accession number GSE230402.

The following datasets were generated:

| Author(s) | Year | Dataset title | Dataset URL | Database and Identifier |
|---|---|---|---|---|
| Cawthorn WP, Ikushima YM, Kobayashi H, Chen K | 2023 | Sex differences in the effects of caloric restriction (CR) on hepatic gene expression in mice | https://www.ncbi.nlm.nih.gov/geo/query/acc.cgi?acc=GSE230402 | NCBI Gene Expression Omnibus, GSE230402 |
| Cawthorn W, Thomas B, Suchacki K | 2022 | Dataset: The effects of caloric restriction on adipose tissue and metabolic health are sex- and age-dependent, 2003-2022 | https://doi.org/10.7488/ds/3758 | Edinburgh DataShare, 10.7488/ds/3758 |

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
