## [Editor Report]

This manuscript breaks new ground in old soil; i.e. sex differences in mouse and human studies and one of the greatest challenges facing translational investigators is the remarkable difference in phenotypes by sex. Defining those differences has been relatively straightforward, but understanding the underlying basis for the phenotypic effects continues to be difficult. This paper demonstrates the sex-and age-dependent effects of calorie restriction on adipose tissue and body composition. It then goes further in describing the metabolic distinctions that arise between males and females. As such, this study will open up new opportunities to explore these sex differences and to test the hypothesis that estrogen in younger animals and humans may be at the center of these effects. Furthermore, it provides a basis for determining why weight loss may be difficult for some individuals with calorie-restricted diets.

---

## [Decision Letter]

[Editors' note: this paper was reviewed by Review Commons.]

---

## [Author Response]

General Statements [optional]

In this study we reveal that, in both mice and humans, the metabolic benefits of caloric restriction (CR) are sex- and age-dependent. Through a systematic review of the literature, we show that sex differences have been largely overlooked by previous CR research, a finding that Reviewer 1 highlights as “an important point”. Our results have critical implications for understanding the fundamental biology linking diet and health outcomes, as well as translational strategies to leverage the therapeutic benefits of CR in humans.

We thank the reviewers for their helpful appraisal of our manuscript, which Reviewer 2 highlights as “a very well written paper”. Reviewer 1 emphasised the translational relevance of our findings and commented on the “systematic” nature of our study. They noted that it “was well performed”, “is a valuable and important contribution to the field”, and “will elicit great interest in the scientific and public readership.” Indeed, the importance of sex as a biological variable was the focus of a recent news feature in Nature (https://www.nature.com/articles/d41586-022-02919-x), underscoring the timeliness and relevance of our findings.

We hope the Editorial Board will agree on the value and importance of our study, which we have further improved through the revisions described below.

Description of the revisions

We have addressed almost all the reviewers’ points through the revisions described in Section 3. In addition, we have conducted extensive new calorimetry studies using the Promethion CORE system, which is the highest-resolution system available for indirect calorimetry. These new data, along with our lipolysis analyses, explain why females resist weight loss and fat loss during caloric restriction.

Description of the revisions that have already been incorporated in the transferred manuscript

We have addressed all of the reviewers’ major comments, as follows:

Reviewer 1Major Comments:A) The clinical part is definitely the weak spot in the study. I don't think that the data should be omitted, but the authors should be very careful in interpreting the data. Obvious limitations apply to this part, which need to be more directly addressed in the abstract and discussion. It feels like the data from the small-scale clinical trial is exaggerated.

The clinical study was conducted by Prof Alex Johnstone’s group at the Rowett Institute of Nutrition and Health, University of Aberdeen. Her group are experts in the study of dietary interventions for weight loss. The study was conducted to a high standard and therefore we have the utmost confidence in the conclusions drawn from our analysis of this data.

As we discuss in response to the reviewer’s other points below, the clinical study was primarily designed to address other outcomes and we analysed the data retrospectively to investigate if sex and age affect CR-induced weight and fat loss. This explains some of the limitations that the reviewer mentions, e.g. the relatively low numbers of younger males, and the focus on overweight and obese subjects. As requested, we have now addressed these limitations as follows:

i. Updated the abstract to clarify that the data are from overweight and obese subjects.

ii. Updated the results to emphasise that we did a retrospective analysis of CR in overweight and obese subjects (lines 447-448).

iii. Performed an additional ANCOVA analysis to test if baseline adiposity or BMI contribute to the sex differences in body mass, fat mass or fat-free mass (new Figure 10—figure supplement 1); see Reviewer 1 Major Point D below.

iv. Updated the ‘Limitations’ section of the Discussion to highlight the retrospective nature of the human study (lines 768-770).

v. Updated the Methods to again clarify the retrospective nature of the analysis (lines 865-867).

B) It is important to mention in the abstract and the discussion that the human data came from obese participants. This might well influence the findings from human data.

The human subjects were overweight or obese; this was previously stated in the methods section (line 867) and in the discussion (lines 554-556). To further clarify this, we now also mention it in the Abstract (line 62) and have reiterated it in the Discussion (line 766-767). Importantly, the fact that humans still show age-dependent sex differences in fat loss, even when overweight and obese, supports our conclusion that this age effect in mice is not simply a consequence of aged mice being fatter than younger mice. We refer to this as the ‘baseline adiposity’ hypothesis (lines 546-563 of the Discussion). In response to point D below, we have also analysed if the loss of fat mass or fat-free mass is influenced by adiposity or BMI at baseline (pre-CR). Our analyses show that neither of these parameters explain the sex differences in loss of fat mass or fat-free mass (see new Figure 10—figure supplement 1).

C) It is very important to calculate the % calorie restriction of the human participants achieved throughout the CR study. This is crucial information to compare it to other studies.

We have updated the Methods (lines 885-889) to explain the basis for the weight loss diet, as follows: “Participants had their basal energy requirements determined and each participant was then fed an individualised diet with a caloric content equivalent to 100% of their resting metabolic rate (Table 3). This approach was taken to standardise the diet to account for individual energy requirements and energy restriction.” We have also updated Table 3 to show the caloric intake for males and females. Note that RMR accounts for ~60-70% of total daily energy expenditure (TDEE) in adults (Martin et al., 2022), so the diet in our study would give a daily caloric deficit of around 30-40% from baseline TDEE.

D) Since there is quite a wide range in the BMIs of the participants, can the authors also stratify against BMI?

We have done this against both baseline BMI and against baseline fat mass (the latter to further test the ‘baseline adiposity’ hypothesis). We now present this data in Figure 10—figure supplement 1. We find that, in males but not in females, baseline BMI or fat mass are significantly associated with the changes in fat mass or fat-free mass: surprisingly, individuals with higher baseline fat mass or BMI show less fat loss and a greater loss of fat-free mass during CR. Importantly, males and females do not significantly differ in the relationships between baseline fat mass (or BMI) and loss of fat mass or fat-free mass. This further supports our conclusion that the sex differences in fat loss are unrelated to differences in baseline adiposity. We report this in lines 445-446 of the Results and lines 558-560 of the Discussion.

E) There is no mention of any pre-study registration online of the clinical part (e.g. clinicaltrials.gov). Was this done?

This study was done before pre-registration was a requirement for clinical trials. We retrospectively analysed the study data to investigate if sex and/or age influence the outcomes. In the updated manuscript we now state this on lines 865-867 of the Methods, as well as in the Results (line 447) and Discussion (lines 768-770).

F) In the methods section the authors write "Participants were informed that the study was funded by an external commercial sponsor…". This is important information, and this is not mentioned anywhere else in the paper. Can the authors clarify this point? A commercial sponsor would, in my view, qualify for a conflict of interest that needs to be mentioned.

We have updated the Declaration of Interests section to clarify this as follows: “The human weight loss study was funded by a food retailer; however, the company had no role in the data analysis, interpretation or conclusions presented in this paper.”

G) How did the authors determine the group sizes for the clinical part? I have some doubts about the sub-group sizes. It would be valuable information if the authors had a statistical analysis plan prior conducting the study. It appears a bit, like the sub-groups were chosen at random, to match findings of the mouse data. Otherwise, there should have been a better allocation within the sub-groups (especially age).

We agree that larger group sizes would have been preferable. This limitation reflects that the study was not originally designed to test age and sex effects on CR outcomes, but instead was analysed retrospectively to investigate the impact of these variables. As mentioned above, we have updated the text of the manuscript to highlight the retrospective nature of the analyses. In the Discussion, under ‘Limitations’, we also highlight the fact that relatively few younger subjects are included in the human study (line 767).

H) There's a big problem with the age stratification of the male participants in the clinical data. If I'm correct there are only 5 males <45 years. Although this looks intriguing, this can easily be a sampling problem.

As above, we agree on the limitation of having relatively few males in this <45 group. We used 45 years as the cut-off point because this is the age when, in women, oestrogen levels begin to decline (as was stated in lines 537-539 of the Discussion, and now reiterated in lines 464-465 of the Results). The key data are those for the ANCOVA analysis (Figure 10H; previously Figure 9H) showing that the influence of age on fat loss differs between males and females across all ages; hence, this is not simply an artefact of the <45 vs >45 groupings.

I) The applied protocol for CR in mice is known to provoke long fasting phases and probably elicits some effects through fasting alone, rather than the caloric deficit. There are some papers out addressing this (e.g. by deCabo, Lamming). The authors should not dismiss this fact and at least address it in their discussion. Also, given this fact, it would be thoughtful to include a database-search – not only regarding CR – but also regarding various types of intermittent fasting protocols in humans and animal studies (similar to what the authors did in the supplemental figure).

We agree on the importance of highlighting recent studies demonstrating that prolonged daily fasting contributes to the outcomes of typical ‘single-ration’ CR protocols. We have added a new paragraph to the Discussion to address this (Lines 742-751).

Regarding the second point, we feel that including a new literature search that addresses not only CR, but also intermittent fasting, is beyond the scope of the current manuscript. However, this is a very good idea and would be worth addressing in a future standalone review article. We have also updated our source data to include all data from our literature reviews, to help if other researchers wish to analyse according to fasting duration or other variables.

J) Did the authors monitor the eating time of the mice?

We have since done this in new cohorts of mice fed using the same CR protocol. We find that the mice consume their food within 2-3 hours, consistent with other CR studies. We have now mentioned this in the Methods section (lines 848-849).

K) While CR certainly has a lot of health benefits in rodents and humans, it should be advised to raise the cautious note that it may not be beneficial for everyone in the general population. For some groups of people and in some cases (e.g. infectious diseases, pregnancy) even CR with adequate nutritional intake of micro/macronutrients might be disadvantageous. This should be mentioned clearly, as the topic gets more and more "hyped" in public media and online.

We now highlight this important point in the opening paragraph of the introduction (lines 73-75).

L) There is no indication of how the authors dealt with missing data. Statistically this can be very important, especially in cases with a low number of data points.

In the Methods section we previously explained (lines 829-830 that “Mice were excluded from the final analysis only if there were confounding technical issues or pathologies discovered at necropsy).” No data had to be excluded from our human study and we have now stated this in the Methods (line 878). For analyses involving paired or repeated-measures data (e.g. time courses of body mass or blood glucose), if data points were missing or had to be excluded for some mice then we used mixed models for the statistical analysis. We have now updated this information in the ‘Statistical analysis’ section of the Methods (lines 1058-1060). Because of the large numbers of mice used in our studies, analyses remain sufficiently well powered even if some data points were missing or had to be excluded.

M) Key data from qPCR should be followed up by western blots or other means. If this was done and there was no effect, the authors should report this. Also, is there any evidence or the possibility to support these findings regarding pck1 and ppara in human samples?

We have now used RNA sequencing to comprehensively determine how CR and sex influence the hepatic transcriptome. These data are reported in new Figures 7 and Figure 7—figure supplement 1 of the revised manuscript. The data identify extensive sex differences in how CR alters hepatic function. Gene set enrichment analysis shows that CR stimulates oxidative phosphorylation and the TCA cycle in males but not in females, even though, in both sexes, there is increased fatty acid oxidation. Moreover, we find that plasma ketone concentrations, a marker of hepatic acetyl-CoA levels, are increased in females compared to males. Thus, our data suggest that CR males use hepatic acetyl-CoA to support the TCA cycle, whereas, in females, acetyl-CoA accumulates, thereby activating pyruvate carboxylase and stimulating gluconeogenesis. These new data substantially improve our manuscript and highlight unexpected sex differences that may underpin the metabolic and health benefits of CR.

Regarding effects of CR on *PCK1* and *PPARA* expression in human liver samples, no human CR studies have taken liver biopsies for downstream molecular analysis. Recent studies of the GTEx database confirm that hepatic gene expression in humans is highly sexually dimorphic (Oliva et al., 2020). We checked *PCK1* and *PPARA* in the GTEx database and found that, in the liver, each of these transcripts is expressed more highly in females than in males (https://www.gtexportal.org/home/gene/PCK1 and https://www.gtexportal.org/home/gene/PPARA). While this is the opposite to what we observe in our ad libitum mice (previous Figure 6A; data now superseded by the RNAseq data), it demonstrates that sex differences in these genes’ hepatic expression do occur in humans. The effect of CR on their hepatic expression, and whether this differs between males and females, remains to be addressed.

N): I think it would be very valuable to analyse the sex-differences in lipolysis directly in fat tissues. The authors concentrated on differences in hepatic mRNA profiles, but there's an obvious possibility and gap in their story.

We agree and have now analysed lipolysis in two ways: firstly, by measuring plasma non-esterified fatty acids (NEFA), and secondly by measuring phosphorylation of hormone-sensitive lipase (HSL) in adipose tissue. These data, shown in new Figures 2K-L and Figure 2—figure supplement 3C, confirm that, during caloric restriction, lipolysis is less active in females than in males, which likely contributes to females’ resistance to fat loss. In the Discussion we cite previous research identifying sex differences in adipose lipolysis and lipogenesis and explain how this data fits with our findings (lines 603-609).

O) Given the relatively low n and sometimes small effect sizes I fear that some of their findings won't be reproduced by other labs. Were the (mouse) data collected all at once in one cohort or did the authors pool data from different cohorts/repeats?

We presume the reviewer means ‘relatively high n’, as most of our mouse analyses used large group sizes. The mouse data were pooled from across multiple cohorts, with ANOVA confirming that the same sex-dependent CR effects were observed within each cohort. This reproducibility across multiple cohorts is a clear strength of our study because it demonstrates the robustness of our findings. Importantly, the sex differences in fat loss, weight loss and glucose homeostasis were still observed in our much-smaller cohort of evening-fed mice (Figure 2—figure supplement 4 and Figure 4—figure supplement 1; n = 5-6), demonstrating that large sample sizes are not needed for other researchers to detect these effects.

Reviewer 1Minor comments:a) The discussion is very extensive, and I suggest compressing the information presented there to make it more easily readable.

We have removed some text that was more speculative, such as the paragraph discussing a possible role for ERalpha. Because of the improved indirect calorimetry analysis, we have also removed the paragraph, under “Limitations”, that explained the limitations of the previous calorimetry studies. We have also revised wording elsewhere to state things more succinctly. However, given the scope of our study we feel we cannot substantially cut down the Discussion without compromising the interpretation of our findings. We note the Reviewer two’s comment that “This is a very well written paper” and feel that attempting to compress the extensive information in the Discussion would compromise, rather than help, the readability.

b) There is some confusion present in the literature regarding the nomenclature of CR/fasting interventions. Recently some reviews have summarized the different forms (e.g. Longo Nature Aging, Hofer Embo Mol Med, …) and the authors should address this briefly. Especially the applied CR intervention in mice overlaps with intermittent fasting.

We have updated the Discussion (lines 742-751) to explain how our single-ration CR protocol also incurs a prolonged intermittent fast, and how this fast *per se* may contribute to metabolic effects.

e) What was the decision basis for stratifying the human data into < and >45 years?

We used 45 years as the cut-off point because this is the age when, in women, oestrogen levels begin to decline (this point was stated in lines 537-538 of the Discussion, and we now reiterate it in lines 464-465 of the Results).

f) The part on aging starting in Figure 7 comes quite surprising and it is not clearly linked to the data before. A suggestion here would be to smooth the transition in the text and the authors could again perform a literature search regarding age-of-onset for CR/fasting interventions in mice and humans.

We have added a sentence to smooth the transition to these studies (lines 415-416), linking the rationale to findings from the RNAseq data that is shown in the previous section. We had previously done a literature search to identify the age of onset of CR interventions in mice and humans. We summarise the findings of this search in lines 502-518 and 531-541 of the Discussion. We have also updated the source data so that it includes our review of the CR literature, allowing other researchers to interrogate this data.

g) At the first mention of HOMA and Matsuda indices, the effect direction should be put into physiological context.

We now mention this in lines 290-291 of the Results.

h) There is no mention of how the PCA analyses were conducted.

We have updated the Methods to explain that the PCA analyses were done using R. We have updated the source data to include the outputs from these analyses, as well as the underlying code. These data and code are now available here https://doi.org/10.7488/ds/3817.

i) Were the mice aged in-house in the authors' facility or bought pre-aged from a vendor? Is it known how they were raised? If bought pre-aged, were female and male animals comparable?

We bred and aged all mice in house. Males and females were littermates from across several cohorts. Therefore, there are no concerns about lack of comparability resulting from environmental differences.

j) Very minor note: I think that "focussed" has become very rarely used, even in British English. I don't know about the journal's language standards, but I would switch to the much more common "focused".

We have updated to ‘focused’ as requested.

k) Figure 6B/F (PCAs) should indicate the % difference of each dimension.

We have updated the figures (now Figure 6D) to show the % variance accounted for by each principal component; this has also been done for the RNAseq data in new Figure 7A. We have also updated the figure legends to specify this.

l) Limitations section: Maybe tone down on "world-leading mass spec facility". This sounds like an excuse and this statement is unsupported and doesn't add anything valuable to the section. Other limitations would include the low n, as mentioned above and the mono-centric fashion of the mouse and human experiments.

We have addressed these points as follows:

– Toned down the description of our mass spec facility (they are renowned for expertise in steroid hormone analysis, so we our original text was intended to highlight that our facility are not novices for this).

– Regarding the low n for some of the human groups, we now highlight this on line 767 of the Discussion.

– We have added a new paragraph to the Discussion (lines 742-751) explaining the limitations of our CR protocol, i.e. that includes elements of both CR and intermittent fasting.

Reviewer 2:Point 1: This is a very well written paper.

We thank the reviewer for this kind comment.

Point 2: Since the authors fed the animals in the morning, this is likely the reason for energy expenditure to be different in the CR vs ad lib groups. Although the authors do study the effects of night v day feeding and saw no change in the outcomes regarding weight, this fact I think should be mentioned somewhere. Also, figure 4A is expressed a W while all the other graphs are in kJ. I think it would be nice to see it all consistent.

Regarding the first point, we agree that time of feeding can influence when energy expenditure is altered, but most studies show that CR decreases overall energy expenditure regardless of time of feeding. For example, Dionne et al. studied the effects of CR on energy expenditure, administering the CR diet during the night phase (Dionne et al., 2016). They found that CR mice have lower energy expenditure in the day but not in the night (Figure 3C in their paper), which is the opposite to our findings (previous Figure 4C; new figures 3A-B). However, total energy expenditure in their study remains decreased with CR. This goes against the reviewer’s suggestion that feeding the animals in the morning “is likely the reason for energy expenditure to be different in the CR vs ad lib groups”. We have updated our manuscript (Lines 611-616) to clarify this.

Regarding the second point, in our new figure 3 we use consistent units (kcal, or kcal/h) for all energy expenditure data. The figure legend also reflects this.

Point 3: For all the graphs, can you make the CR groups bold and not filled as it is hard to see the lighter colours.

We have updated the graphs so that the CR groups are represented by solid lines, rather than dashed lines.

Point 4: I know many investigators use them, but I am not sure how relevant HOMA-IR and the Matsuda index are in mice since they were specifically designed for humans.

The issue of whether it is correct to use HOMA-IR and/or Matsuda index in mice is often debated in the metabolism field. Importantly, we are not using the absolute values for HOMA-IR or Matsuda in the same way that they are used in humans; instead, we are comparing the relative values between groups because these are still physiologically meaningful. We discussed this with Dr Sam Virtue, an expert in mouse metabolic phenotyping (Virtue and Vidal-Puig, 2021), who agrees on their usefulness in this way.

Point 5: Something also to note is the fact that all the glucose uptake data is under basal conditions. Just because there are no differences in the basal state does not mean that there are no differences after a meal/during an insulin stimulation. I think that this needs to be discussed and the muscle and fat not completely discounted as a player in the differences seen.

We agree that CR can enhance insulin-stimulated glucose uptake, but our OGTT data suggest that it is effects on fasting glucose, rather than insulin-stimulated glucose uptake, that contribute to the sex differences we observe. We have now updated the Discussion (lines 642-645) as follows, “CR enhances insulin-stimulated glucose uptake (83) and it is possible that this effect differs between the sexes. Our second relevant finding is that, during an OGTT, CR decreases the tAUC but not the iAUC, highlighting decreases in fasting glucose, rather than insulin-stimulated glucose disposal, as the main driver of the improvements in glucose tolerance.”

Description of analyses that authors prefer not to carry outReviewer 1:Major point I: … it would be thoughtful to include a database-search – not only regarding CR – but also regarding various types of intermittent fasting protocols in humans and animal studies (similar to what the authors did in the supplemental figure).

We feel that including a new literature search that addresses not only CR, but also intermittent fasting, is beyond the scope of the current manuscript. However, this is a very good idea and would be worth addressing in a future standalone review article. To assist with this, we have updated the source data to include all details of the literature review presented in Figure 1. The link to the source data is provided in the manuscript.

Minor point c: The order of the subpanels in Figure 9 (and other figures where B is below A and so on) is confusing. Please rearrange or indicate in a visual way which panels belong to each other.

We disagree that the order of subpanels in Figure 9 (now Figure 10) is confusing: the panels are clearly labelled, and we find it most logical to have the absolute values shown in the top row (panels A, C and E), with the corresponding graphs of fold changes shown beneath each of these (panels B, D and F). This allows the reader to quickly compare the absolute vs fold-change data for each readout. If we had panels A-C on the top row and D-F on the second row, then the connection between graphs 10C and 10D would be less clear and comparable.

Minor point d: Did the authors also measure cardiovascular (e.g. blood pressure) parameters? There is some evidence out there that there is an age/sex dependency during fasting/CR. This would be a nice add-on to the rather small clinical data here.

We did measure various cardiovascular parameters for our mice but find, unlike for the metabolic outcomes, these generally don’t show sex or age differences. In our human study we measured blood pressure and heart rate before starting CR and at weeks 3 and 4 post-CR. For this response to reviewers we have summarized these human data in Author response image 1. The data show that CR decreases blood pressure and heart rate in males and females (Author response image 1). In the younger age group (<45 years) the decrease in systolic blood pressure is greater in females than in males (Author response image 1) and the changes in heart rate over time also differ between the sexes, irrespective of age (Author response image 1). However, unlike for the effects on fat mass or fat-free mass (now shown in Figure 10H-I; previously Figure 9), across all subjects ANCOVA reveals no age-sex interactions in these cardiovascular effects.

We have decided to not include these data in the current study because we feel it is already extensive and is focused on metabolic outcomes. We instead plan to report the cardiovascular outcomes (from both humans and mice) in a separate paper.

**Author response image 1. sa2fig1:** Effects of CR on cardiovascular parameters in humans (related to Figure 10). Twenty male and twenty-two female volunteers participated in a weight loss study involving a 4-week dietary intervention, as described for Figure 9. (A-F) Systolic blood pressure (BP), diastolic BP and heart rate were recorded at weeks 0, 3 and 4. Data are shown as absolute values (A,C,E) or fold-change relative to baseline (B,D,F). Data are presented as mean ± SEM. Significant effects of time, sex, and time*sex interaction were assessed using 2-way ANOVA. (G-I) Simple linear regression of age vs fold-change (*week 4 vs week 0)* in systolic BP (G), diastolic BP (H) and heart rate (I). For each sex, significant associations between age and outcome (fold-change) are indicated beneath each graph as ‘*P,* Slope’. ANCOVA was further used to test if the age-outcome relationship differs significantly between males and females. ANCOVA results are reported beneath each graph as ‘*P,* Slope’ and ‘*P,* Intercept’ for males vs females (M vs F). In (G), similar slopes but different intercepts show that sex significantly influences changes in systolic BP, but the influence of age does not differ between the sexes. In (H,I) neither slopes nor intercepts differ significantly between males and females, indicating that the age-outcome relationship is similar between the sexes. (J-L) Fold-change (week 4 vs week 0) in systolic BP (J), diastolic BP (K) and heart rate (L) for males vs females separated into younger (<45 years) and older (>45 years) groups. Data are presented as violin plots overlaid with individual data points. Significant effects of age, sex, and age*sex interaction were assessed using 2-way ANOVA with Tukey’s multiple comparisons test. Overall *P* values for each variable, and their interactions, are shown beneath each graph. Significant differences between comparable groups are indicated by ** (*P*<0.01).

References:

Dionne, D.A., Skovso, S., Templeman, N.M., Clee, S.M., and Johnson, J.D. (2016). Caloric Restriction Paradoxically Increases Adiposity in Mice With Genetically Reduced Insulin. Endocrinology *157*, 2724-2734. 10.1210/en.2016-1102.

Martin, A., Fox, D., Murphy, C.A., Hofmann, H., and Koehler, K. (2022). Tissue losses and metabolic adaptations both contribute to the reduction in resting metabolic rate following weight loss. Int. J. Obes. *46*, 1168-1175. 10.1038/s41366-022-01090-7.

Oliva, M., Muñoz-Aguirre, M., Kim-Hellmuth, S., Wucher, V., Gewirtz, A.D.H., Cotter, D.J., Parsana, P., Kasela, S., Balliu, B., Viñuela, A., et al. (2020). The impact of sex on gene expression across human tissues. Science *369*, eaba3066. 10.1126/science.aba3066.

Virtue, S., and Vidal-Puig, A. (2021). GTTs and ITTs in mice: simple tests, complex answers. Nat Metab *3*, 883-886. 10.1038/s42255-021-00414-7.